# TLINet: A defects detection method for insulators of overhead transmission lines using partially transformer block

Xun Li[1,2,3], Yuzhen Zhao[1]*, Yang Zhao[1], Zhun Guo[1], Yongming Zhang[1], Xiangke Jiao[1], Baoxi Yuan[1,2,3]

**1** Xi'an Key Laboratory of Advanced Photo-electronics Materials and Energy Conversion Device, School of Electronic Information, Xijing University, Xi'an, China, **2** Xi'an Key Laboratory of High Precision Industrial Intelligent Vision Measurement Technology, Xijing University, Xi'an, Shaanxi, China, **3** Xi'an Key Laboratory of Intelligent Sensing and Autonomous Navigation for Low Altitude Vehicles, Xijing University, Xi'an, Shaanxi, China

\* zyz19870226@163.com

## Abstract

The defects of insulators exhibit characteristics such as complex backgrounds, multi-scale variations, and small object sizes. Therefore, accurately focusing on these defects in dynamic and complex natural environments while maintaining inference speed remains a pressing challenge. To address this issue, this paper proposes an innovative insulator defect detection network, TLINet. First, a Multi-Branch Partially Transformer Block (MBPTB) is designed to enhance the backbone's capability in capturing global features. Next, a Dynamic Downsampling Module (DyDown) is introduced to mitigate the issue of small-scale defect information blurring. Furthermore, considering the multi-scale variations of insulator defects, this paper proposes a Context-Guided Feature Fusion Network (CGFFN). This module enables fine-grained fusion of features at different scales, allowing the model to generate adaptive responses to defects of various sizes. Compared to the baseline model, the proposed method improves mAP50 by 5.3% on our self-constructed Insulator-DET dataset. On CPLID-D and CPLID-N, it achieves mAP50-95 improvements of 7.9% and 12.1%, respectively. Additionally, to verify the robustness of the proposed algorithm, TLINet is evaluated on the VOC07 + 12 dataset. Compared to the baseline model, TLINet improves mAP50 by 0.4% while reducing the number of parameters by 1/6. These results demonstrate the effectiveness of TLINet in addressing the complexities of insulator defect detection in power transmission lines. The code is available at https://github.com/mazilishang/TLINet.

## 1. Introduction

Transmission lines serve as the bridge between power plants and electrical loads within the power system, and their safety directly impacts the transmission of

**Data availability statement:** The datasets used in this study are all publicly available on the Kaggle platform, including: Insulator-DET Dataset: https://www.kaggle.com/datasets/mazilishanglx/insulator-det IDID Dataset: https://www.kaggle.com/datasets/mazilishanglx/idid-dataset CPLID Dataset: https://www.kaggle.com/datasets/mazilishanglx/cplid-dcplid-nVOC07+12 Dataset: https://www.kaggle.com/datasets/mazilishanglx/voc07-12 These datasets comprise a variety of image collections used for model training, validation, and evaluation, covering diverse insulator defect images and natural scene images. All relevant data are freely accessible, ensuring the reproducibility of this study.

**Funding:** This work was supported by Natural Science Basic Research Plan in Shaanxi Province of China (No. 2024JC-YBMS-342), Science and technology plan project of Xi'an (No. 24GXFW0091), the Youth Innovation Team of Shaanxi. This work was supported by the Natural Science Foundation of Shaanxi Province [2021JM-537], in part by the Key Program of the National Social Science Foundation of China (NSSFC, 23AGL039), in part by the Shaanxi Provincial Science and Technology Plan Project (2024GX-YBXM-114), and in part by the Natural Science Foundation of Shaanxi Province (Grant No. 2023-YBGY-036).

**Competing interests:** The authors declare that the research was conducted in the absence of any commercial.

electrical energy. As a crucial component of transmission lines, insulators prevent accidents such as short circuits and leakage currents. However, due to prolonged exposure to complex and variable natural environments—such as low temperatures, rain, snow, and heavy fog climatic changes can directly affect the electrical performance and mechanical strength of insulators. If an insulator fails, it can directly lead to power grid failures and even large-scale blackouts. Therefore, accurately and promptly monitoring the health status of insulators is critical to ensuring the safety of power supply [1–4].

Insulator defect detection faces challenges such as large-scale variations, small object sizes, and complex backgrounds. Traditional transmission line inspection methods rely on manual operations, but this approach is time-consuming and struggles to detect issues in a timely manner. In recent years, Unmanned Aerial Vehicles (UAV) have emerged as a primary means for inspecting high-altitude transmission lines due to their flexibility and efficiency. However, as power grids continue to expand, manually identifying defects from images becomes increasingly labor-intensive. While traditional machine vision-based detection methods improve efficiency, the manual feature extraction process adds to the workload [5,6]. As a result, deep learning techniques have been widely applied to insulator defect detection. Deep learning-based insulator defect detection methods can be broadly categorized into two types: one represented by traditional convolutional neural network (CNN) models such as the You Only Look Once (YOLO) series [7–11], and the other represented by Vision Transformer models such as Real-Time Detection Transformer (RT-DETR) [12,13].

To address the key challenges of multi-scale targets, complex backgrounds, and environmental interference in insulator defect detection during high-altitude power transmission line inspections, this paper proposes a multi-scale dynamic feature modeling method that combines convolutional neural networks and Transformers. The specific objectives include designing a dynamic downsampling module to reduce semantic information loss, constructing a multi-branch semantic guidance module to enhance the distinction between defects and background, and integrating the advantages of both network architectures to optimize detection accuracy and inference efficiency. This approach aims to achieve efficient and robust automatic detection of insulator defects, ensuring the safe and stable operation of the power grid.

Existing research indicates that CNN-based detectors excel in extracting local features, achieving fast inference speeds, and maintaining low computational costs [14]. However, they perform poorly in capturing non-local information and suppressing noise interference. In contrast, Transformer-based detectors establish cross-region relationships between any pixel in an image and all other pixels, allowing them to better capture fine-grained differences and overall structural patterns. However, the quadratic complexity of the self-attention mechanism limits the processing efficiency of Transformers in high-resolution UAV aerial images, especially under hardware-constrained UAV platforms [15]. Recent studies [16] have proposed hybrid architectures that integrate Transformers with CNNs. This approach aims to leverage the global feature extraction capability of Transformers while retaining the CNN's

strength in local feature representation, thereby enhancing performance across various vision tasks. However, this combination alone does not fully address the challenge of large-scale variations in insulator defects. When dealing with defects of significantly different scales, CNNs may struggle to cover all scale details due to their local nature, while Transformers, despite providing broad global information, often fail to precisely align with multi-scale defect features.

To bridge this gap, this paper proposes MBPTB. MBPTB enhances the model's responsiveness to multi-scale features by decoupling feature channels. This strategy not only optimizes information flow and reduces detail loss but also significantly improves detection accuracy and efficiency. This innovation enables the CNN-Transformer architecture to better adapt to the demands of multi-scale defect detection, enhancing its practical applicability in high-altitude transmission line inspections.

In the context of insulator defect detection, defects typically appear as small-scale targets, occupying only a limited area in the image, as shown in Fig 1. However, as neural networks progressively downsample through multiple layers, the resolution of input feature maps decreases, leading to a gradual loss of fine details in small-scale defects. Conventional single-path downsampling approaches overly rely on global features while neglecting the differences among features at different scales. Additionally, the fixed convolutional kernel limits the module's ability to adapt to defects of different scales. As a result, when processing objects with significant scale variations, the model often struggles to effectively capture small defect features, leading to information loss. To address this issue, this paper proposes a dynamic downsampling strategy that uses separation and fusion. DyDown enhances the model's ability to capture multi-scale features by partitioning the feature map and processing each segment separately. By integrating max pooling with dynamic convolution, this design overcomes the limitations of fixed convolutional kernels in traditional downsampling modules, providing a more flexible and efficient approach for handling multi-scale objects. Through this innovation, DyDown improves adaptability to features of different scales, making it particularly suitable for insulator defect detection tasks involving significant scale variations.

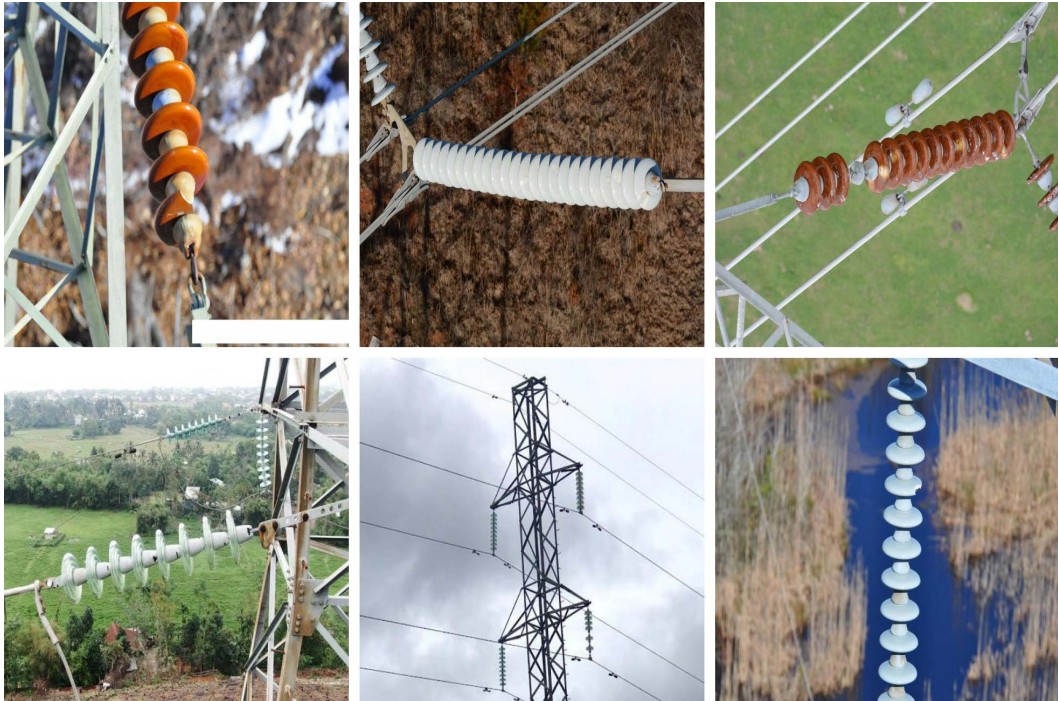

**Fig 1. UAV aerial image samples of insulator.**

The environment surrounding transmission lines is complex, often containing interfering factors in the background that may obscure the characteristics of insulator defects. Accurately distinguishing defects in such complex backgrounds is a major challenge in insulator defect detection. Feature Pyramid Networks (FPN) and Path Aggregation Networks (PAN) enhance feature transmission across different levels, thereby improving detection accuracy for small-scale objects [17,18]. The Bidirectional Feature Pyramid Network (BiFPN) further strengthens bottom-up information flow. Through this bidirectional fusion, BiFPN can better capture multi-scale information and effectively mitigate information loss [19]. However, BiFPN has a relatively complex structure, and its high computational cost limits its widespread deployment on UAV platforms. Moreover, the aforementioned methods mainly focus on feature aggregation and information flow across scales while overlooking the critical role of the surrounding environment in defect localization. Precisely locating defects requires not only understanding their intrinsic characteristics but also comparing them with their surrounding environment. To address these issues, this paper proposes MBCGM. MBCGM extracts features from both the defect itself and its surrounding environment, enhancing the joint representation of multi-scale features while optimizing the contrast between background and object features. By incorporating MBCGM, CGFFN overcomes the limitations of traditional feature fusion networks in handling background interference, providing a more efficient and accurate solution for insulator defect detection in high-altitude transmission line inspections.

To summarize, the main contributions of this paper are as follows:

1. The MBPTB was proposed in this paper. This module enhances the model's ability to respond to multi-scale features by separating feature channels. Additionally, MBPTB can suppress background noise interference, allowing the model to focus on the defect itself.

2. The DyDown was proposed in this paper. This module overcomes the limitations of fixed convolutional kernels and single-path structures in traditional downsampling modules. It introduces a more flexible and efficient multi-path dynamic downsampling strategy.

3. The CGFFN was proposed to optimize traditional feature fusion networks. This network can more accurately handle the feature fusion of defects at different scales, enhancing the model's adaptability to variations in the size and shape of insulator defects.

The remainder of this paper is organized as follows. Section 2 provides an overview of related work on small object detection and multi-scale feature extraction. Section 3 introduces the proposed TLINet and its associated improvement strategies. Section 4 presents the implementation details and analysis of experimental results. Section 5 concludes the paper and discusses future research directions

## 2. Related work

Insulator defect detection is one of the research hotspots in the field of object detection. Insulator defects are characterized by significant scale variations, complex backgrounds, and small target sizes. To address these challenges, many researchers have proposed relevant studies.

### 2.1. Tiny defect detection

Currently, research on insulator defect detection primarily focuses on improving the recognition capability of models for small-scale defects, especially in complex backgrounds and real-time processing scenarios. Wang et al. [20] proposed the MCI-GLA module, which enhances the recognition of small-scale defects by effectively capturing both global contextual information and local details. Although this method significantly improves the detection accuracy of small targets, the MCI-GLA module has a large number of parameters, leading to high computational resource consumption, which limits its application in real-time video processing. Lu et al. [21] introduced the ConvSimCB module, which enhances

the model's ability to focus on defect features in complex backgrounds, thereby significantly improving recall and accuracy. However, due to its high computational complexity, this method is difficult to deploy efficiently in resource-limited environments. Li et al. [22] proposed LiteYOLO-ID, which optimizes the network structure and reduces the number of parameters, effectively improving detection accuracy and speed. Although this method significantly enhances efficiency in small-object detection, it still faces challenges in accurately recognizing defects with complex details. In particular, its recognition capability declines when defect morphology varies significantly. Zhou et al. [23] proposed a two-stage image feature extraction method, which first extracts insulator features from complex backgrounds and then conducts fault analysis. While this approach significantly improves recall, its accuracy may be limited in cluttered backgrounds or occluded scenarios. Ma et al. [24] introduced a visual hierarchical attention detector that improves the detection of small defects through target feedback. Although this method performs well in small-object detection, its ability to recognize defect features may become unstable under significant lighting variations. Yu et al. [25] proposed a bidirectional fusion structure that effectively enhances small-object detection, enabling the model to better focus on small objects. However, while this method enhances precision, it may struggle to accurately locate the object when both large-scale defects and small defects coexist. AdIn-DETR addresses the issues of poor cross-domain generalization and low recognition accuracy under small-sample conditions in power insulator defect detection by introducing a domain-adaptive query module and a foreground-background contrast enhancement mechanism. However, AdIn-DETR suffers from slow inference speed and still exhibits unstable feature extraction under extreme environments, which limits its effectiveness in practical inspection scenarios [26].

To address the aforementioned issues, this paper introduces MBPTB and DyDown, focusing on enhancing edge feature extraction and minimizing detail loss. MBPTB balances local feature extraction with global contextual information acquisition, ensuring accurate recognition and localization of small-scale defects in complex scenes. DyDown employs a unique decoupled and fused downsampling strategy to accurately match small-scale defects with extreme aspect ratios, thereby reducing detail loss during the downsampling process.

## 2.2. Multi-scale feature extraction

In the task of insulator defect detection, the model needs to adapt to variations in defect scale to improve detection accuracy and efficiency. To address this, researchers have proposed various methods in recent years to enhance multi-scale object detection capabilities. Hao et al. [27] proposed the ID-YOLO model, which utilizes a multi-scale bidirectional feature pyramid network to handle small-scale defects. This method performs well in multi-scale defect detection. However, due to its inadequate adaptation to complex backgrounds, it may lead to false positives or missed detections in low-contrast or cluttered scenes, limiting its general applicability in real-time scenarios. Zhang et al. [28] introduced an attention mechanism to improve the detection accuracy of damper defects. Although this method performs well in multi-scale defect detection, its robustness is relatively weak when dealing with high-noise backgrounds, due to the lack of an effective noise suppression strategy. Song et al. [29] adopted multi-scale information feature fusion strategy to enhance the detection of multi-scale objects. While this approach significantly improves accuracy, it does not precisely balance attention across different scales, leading to suboptimal detection performance in cases of highly overlapping objects. Zhang et al. [30] proposed the lightweight DsPAN model to address the detection of small-scale defects. Although this method achieves good accuracy, its ability to capture fine details remains limited in highly variable backgrounds, affecting its stability in real-world applications. Cheng et al. [31] introduced an efficient channel attention module that enhances the feature extraction capability for small and elongated defects using one-dimensional convolution. While this method improves multi-scale defect detection accuracy, it lacks generalization when handling defects of varying sizes and shapes. Particularly in images containing large defects, the model may overlook small objects. Jing et al. [32] proposed a plug-and-play feature aggregation network, which strengthens low-level spatial information through a top-down dual-pathway approach. This method excels in small-object detection, but the introduction of a complex network structure results in high computational

overhead during inference. Tian et al. [33] proposed a Small Object Intersection over Union (SOIoU) loss function based on object feedback, which adaptively optimizes small-object sizes to improve the model's focus on small objects. While this method enhances accuracy in small-object detection, its adjustment mechanism may introduce instability when handling objects with large scale variations. Li et al. [34] designed a Frequent Interaction Feature Fusion Network to address the scale imbalance issue in UAV aerial images. This method enhances interactions among features at different levels through dense skip connections, effectively improving the model's robustness to scale variations. However, the dense skip connections increase computational complexity, resulting in slower inference speed.

To address the aforementioned challenges, the CGFFN was proposed to more precisely handle feature fusion for defects of different scales and enhance the network's adaptability to variations in insulator defect size and shape.

## 2.3. Background noise removal

In insulator images, there are background objects that resemble the color or texture of insulators, which significantly affecting the model's detection performance. To address this issue, many researchers have proposed different background noise suppression methods. Yang et al. [35] balanced detection accuracy and inference speed through bidirectional information fusion. However, despite the accuracy improvement, the model still lacks robustness when dealing with high-noise backgrounds, limiting its widespread application in real-time tasks. Panigrahy et al. [36] mitigated the impact of complex backgrounds by employing various image augmentation techniques, effectively reducing overfitting. However, this method relies heavily on artificial image augmentation, and these augmentation strategies may fail to adapt to dynamically changing environments. Luo et al. [37] proposed the Ultra-Small Bolt Defect Detection Model (UBDDM) based on a global-local detection framework. This model utilizes a self-attention mechanism to improve detection accuracy for small objects, particularly in complex backgrounds. Although this method achieves good accuracy, it still exhibits a high false positive rate in images with complex and frequently changing backgrounds, making it challenging to meet real-time detection requirements in dynamic environments. Fu et al. [38] enhanced the accuracy of small object detection in complex backgrounds through hierarchical feature fusion. However, the method suffers from redundant computations during feature fusion, leading to unstable detection accuracy in extreme scenarios. Yuan et al. [39] designed a Multi-Convolution Block Attention Module (MCBAM) to enhance the ability to extract defect features from complex backgrounds. Although this module effectively improves the model's adaptability to complex backgrounds, it introduces additional model complexity, resulting in lower processing efficiency when handling multiple small objects simultaneously. Zhou et al. [40] enhanced the integration of semantic information by introducing a bidirectional fusion strategy, thereby improving the contrast between defects and the background; however, detection accuracy may still decline under complex lighting conditions. Tie et al. [41] optimized the feature representation of the detection head through attention mechanisms, enabling the detector to accurately learn the relationship between steel defects and the background. Nevertheless, this method primarily focuses on feature representation optimization and fails to provide stable detection performance in environments with complex noise.

To address these challenges, the MBPTB was proposed to suppress background noise. In addition, MBPTB can enhance the model's perception of global features while maintaining computational efficiency.

## 3. Method

Based on YOLOv8n, the TLINet was proposed to address the challenges in insulator defect detection for transmission lines. TLINet introduces three key innovations: MBPTB, DyDown, and CGFFN. MBPTB captures rich and valuable high-level semantic information along with high-resolution fine-grained details, enabling the model to focus on objects within complex and dynamic backgrounds. CGFFN consists of the proposed MBCGM and C2f modules. MBCGM dynamically responds to different input features, enhancing the multi-scale representation capability of the feature fusion network. DyDown reduces the spatial resolution of feature maps while preserving as much image information as possible, allowing

the model to capture multi-scale features at higher levels. Apart from these improvements, the rest of TLINet retains the original YOLOv8n network structure. The detailed network architecture of TLINet is shown in Fig 2.

### 3.1. Muti-branch partially transformer block

During drone flights, variations in altitude and angle significantly affect the background composition of captured images, leading to increased complexity and diversity in the backgrounds of insulator images. Moreover, differences in shooting distance cause drastic changes in defect scales, placing higher demands on the model's multi-scale modeling capability. To balance local and global information modeling while ensuring robustness, this paper proposes a multi-branch partial Transformer block.

Compared to traditional CNNs, which excel mainly at local modeling, Transformers possess powerful global perception abilities; however, their high computational cost makes deployment on resource-constrained platforms like drones challenging. To balance global modeling capacity and computational overhead, the input feature map is divided into two parts and processed in parallel by a lightweight CNN module and a Transformer module, forming a hybrid structure called the Partially Transformer Block (PTB). This structure enables cooperative extraction of local and global features, preserving CNN's strengths in capturing textures and edges as local details, while leveraging the Transformer's ability to model long-range dependencies, effectively mitigating the interference of background noise on feature extraction.

However, the original PTB structure has limitations when facing defects of varying scales, as its fixed feature processing paths cannot fully integrate multi-scale feature information. To enhance the model's adaptability in multi-scale scenarios, this paper further designs the MBPTB by introducing a multi-branch feature processing strategy to increase PTB's flexibility and generalization.

The detailed architecture of MBPTB is shown in Fig 3. Given the input feature map $X_i \in \mathbb{R}^{H \times W \times C\_in}$, a 1 × 1 convolution is first applied to reorganize features and adjust the channel dimension to reduce computational complexity. Then, the features are split into two branches: an identity mapping branch for information preservation and gradient flow, and a PTB

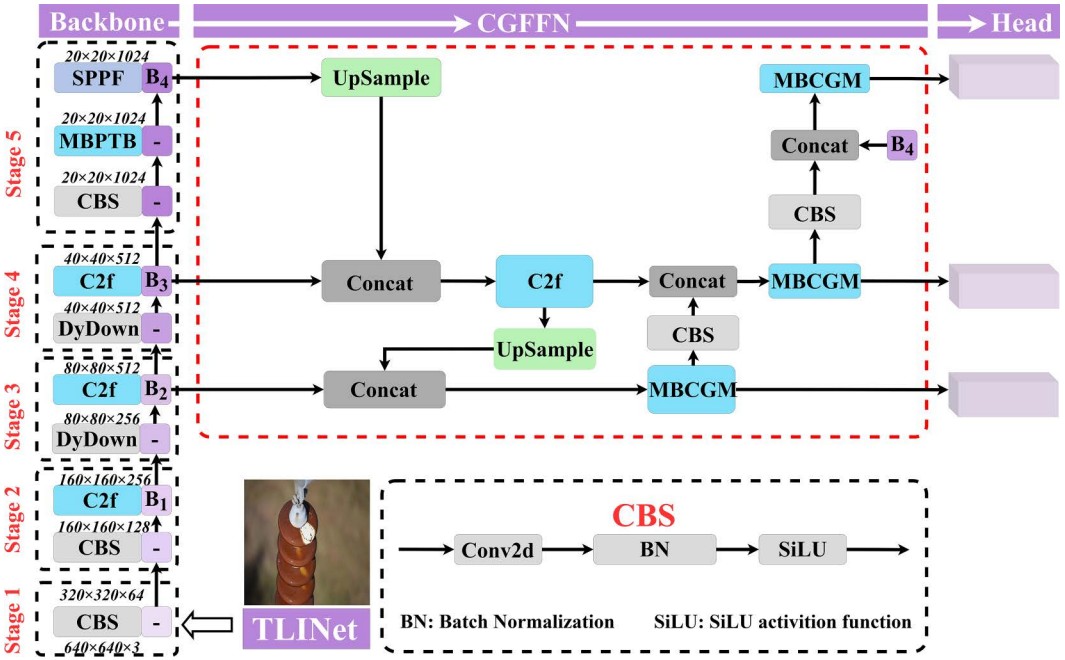

**Fig 2. General framework of the proposed TLINet. The red dashed box represents the proposed CGFFN.**

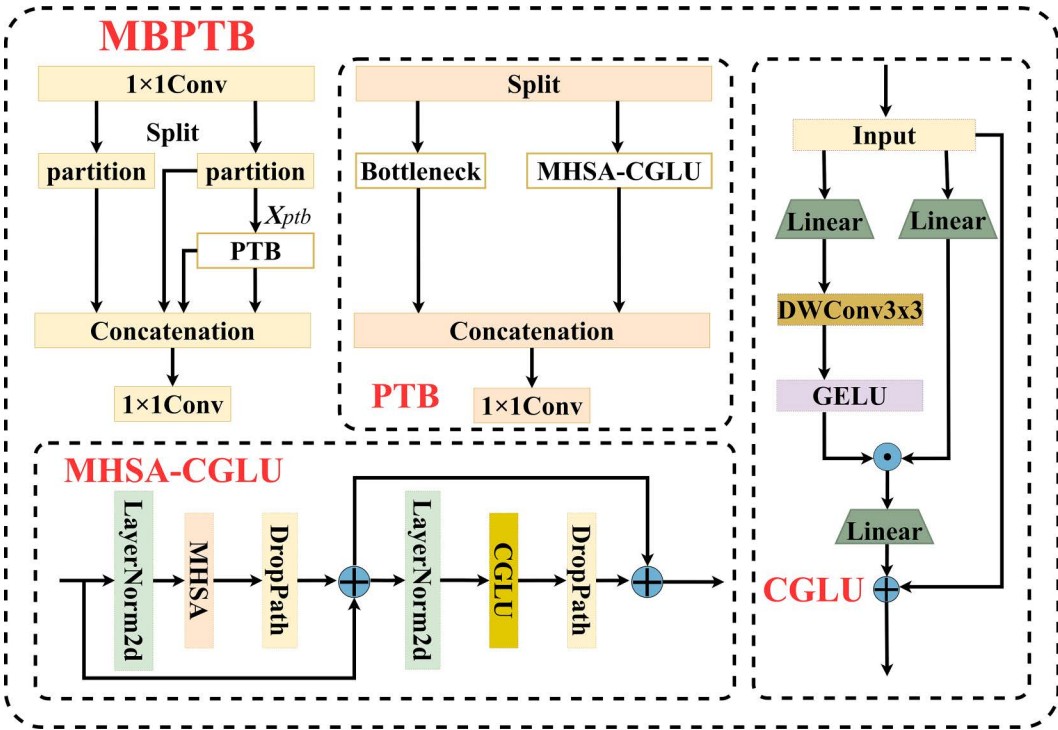

**Fig 3. Structure of Multi-Branch Partially Transformer Block.**

branch to enhance feature representation. The PTB branch is further divided into two paths: a Bottleneck branch and an MHSA-CGLU branch, which are responsible for extracting local details and modeling global relationships respectively, forming a complementary local-global modeling structure. The working principle of MHSA is described as follows:

$$MHSA = Concat(head_1, \ldots, head_i)N^o \tag{1}$$

$$head_i = ATT(Q_i, K_i, V_i) = softmax(\frac{Q_i K_i^T}{\sqrt{d_k/h}})V_i \tag{2}$$

Where *MHSA* represents the multi-head self-attention module, *Concat* denotes the concatenation operation, and $N^o \in R^{d_k \times d}$ is the learnable weight matrix. $head_i$ represents the output of the *i*-th attention head. $Q_i, K_i, V_i$ denote the query, key, and value vectors of the *i*-th head, respectively. The *softmax* function is used to compute the attention scores, which measure the correlation between the query and key vectors. The scaling factor $\sqrt{d_k}$ is introduced to stabilize training. *h* denotes the number of attention heads, which is set to 8 in MBPTB.

Specifically, the Convolutional Gated Linear Unit (CGLU), by introducing a gating mechanism, can dynamically adjust the importance of information at different scales, enabling more precise and efficient multi-scale feature fusion [42]. Through its parallel structure design, MBPTB demonstrates superior robustness and feature discrimination capability in scenarios involving varying scales, complex textures, and background noise. Additionally, MBPTB divides the feature map and processes it along different paths, effectively avoiding the redundant computations caused by directly transmitting large-scale feature maps, thus achieving a balance between model efficiency and performance.

Compared to single modules, MBPTB combines the lightweight nature of CNNs, the global modeling ability of Transformers, and the dynamic feature regulation of CGLU, forming a structurally sound, highly expressive, and computationally manageable high-performance feature extraction unit. This module provides a solid feature foundation for identifying small defects under multi-scale and complex background conditions throughout the detection network.

### 3.2. Muti-branch context-guided module

In the context of insulator defect detection, defects typically exhibit characteristics such as small size, blurred edges, and weak visual features, often appearing in regions with complex background interference. These challenges hinder traditional feature extraction and fusion strategies, particularly in handling complex environments and multi-scale targets. Feature fusion modules play a critical role in integrating shallow fine-grained details with deep semantic information. However, existing approaches often rely on fixed or single-scale feature aggregation strategies, overlooking the contextual information surrounding defects and the synergy between features at different scales. As a result, they struggle to effectively model the relationship between local defects and their surrounding context [43].

To address these issues, this study conducts an in-depth analysis of the structural characteristics of insulator defect images and proposes the MBCGM module, which serves as the foundation for constructing the CGFFN. CGFFN incorporates the following key design mechanisms to better handle complex backgrounds and multi-scale defect detection tasks.

First, the MBCGM module adopts a path separation strategy that explicitly distinguishes between the "defect itself" and its "surrounding context" for feature modeling. Its dual-path structure independently processes local detail information and background semantic information, enabling the network to focus not only on the target area but also to perceive the contextual surroundings. This enhances the network's ability to distinguish defects from complex backgrounds. Next, the embedded CGBlock within MBCGM integrates both standard convolution and dilated convolution branches to capture local details and contextual information with a larger receptive field, respectively. This design addresses the susceptibility of small target detection to background interference and enhances the model's multi-scale perception capabilities. Additionally, the average pooling branch in CGBlock introduces global context to guide the weight distribution of local feature branches, allowing the network to dynamically emphasize target-relevant regions while suppressing irrelevant background features, thereby improving defect localization accuracy. Finally, an identity mapping branch retains the original input information to prevent the loss of critical details during fusion. The final output feature map is formed by concatenating outputs from multiple branches, further enriching feature diversity and representation capacity.

In summary, CGFFN significantly improves the robustness and accuracy of multi-scale defect detection in complex backgrounds by integrating mechanisms such as context guidance, multi-scale perception, global regulation, and residual information preservation. It achieves a well-balanced trade-off between detection precision and computational efficiency.

Fig 4 illustrates the detailed structure of MBCGM. Suppose $F_i \in \mathbb{R}^{H \times W \times C\_in}$ is the input feature map of MBCGM, where $H$ and $W$ represent the height and width of the feature map, respectively, and $C\_in$ is the number of input channels. First, $F_i$ is fed into the CBS module. At this stage, the size of the feature map is $H \times W \times C\_out$. Subsequently, the feature map is evenly divided into two parts, which are then input into the main branch and the identity mapping branch. The main branch is further divided into the identity mapping branch and the CGBlock branch. At this stage, the size of the feature maps for both branches are $H \times W \times C_{out}/2$. The CGBlock branch consists of two sub-branches: the first is a conventional 3×3 convolution branch, while the second is a 3×3 dilated convolution branch. Through the path separation strategy, the model can independently extract both the fine-grained features of the defect and the contextual features of its surrounding environment.

Next, the output feature maps of the two CGBlock's sub-branches are concatenated, forming a joint feature map that integrates both local details and contextual information. The CGBlock is then divided into an identity mapping branch and an average pooling branch. The global information obtained from the average pooling branch is used to weight the feature map input into the Fusion module, preserving crucial local information while enhancing the influence of global information

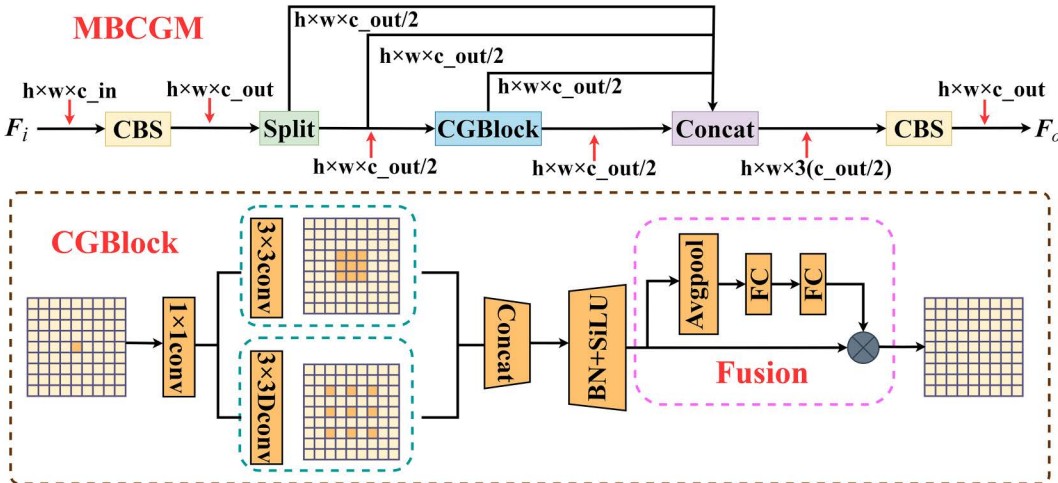

**Fig 4. Architecture of Multi-branch Context-Guidance Module.**

on local features. Meanwhile, the identity mapping branch ensures that the original information is not excessively compressed or lost. These two branches are fused through element-wise multiplication. The integration of global and local features effectively enhances the model's ability to comprehend multi-scale features.

Finally, the feature maps from the CGBlock and identity mapping branch are concatenated. At this stage, the size of the feature map is $H \times W \times 3(C_{out}/2)$. This feature map is then passed through a CBS module to obtain the MBCGM's output feature map $F_o \in \mathbb{R}^{H \times W \times C\_out}$. It is worth noting that each CBS module consists of a cascaded 3×3 convolution module, a batch normalization layer, and a SiLU activation function.

### 3.3. Dynamic downsample module

In insulator defect images, targets are typically small in size and exhibit significant scale variation. Traditional downsampling methods, such as fixed convolution kernels or single-path pooling operations, although effective in reducing spatial resolution and computational complexity, often sacrifice fine-grained features. This is especially problematic when dealing with small-scale defects, as it can lead to information loss and reduced detection accuracy [44]. To address this issue, this paper proposes DyDown, which aims to retain critical detail information while maintaining computational efficiency. Unlike traditional approaches, DyDown adopts a multi-path structure and incorporates a dynamic convolution mechanism, allowing the convolutional kernels to adaptively adjust based on the input features, thereby better capturing defect features at various scales. Specifically, DyDown first applies average pooling to compress the input feature maps and enhance the retention of local details. The pooled features are then evenly split into two branches: one employs dynamic convolution to improve adaptability to scale variation, while the other combines max pooling with dynamic convolution to highlight salient regions and suppress background noise. Finally, the features from both branches are concatenated to effectively integrate global semantics and local details. This structure not only enhances multi-scale feature representation but also improves the perception of small defect targets while reducing redundant computation, significantly outperforming traditional fixed-structure downsampling modules.

Fig 5 illustrates the detailed structure of DyDown. Let $H_i \in \mathbb{R}^{H \times W \times C}$ be the input feature map of DyDown, where $H$ and $W$ represent the height and width of the feature map, respectively, and $C$ denotes the number of input channels. First, $H_i$ is passed through a 2×2 average pooling layer, resulting in $H_j \in \mathbb{R}^{H \times W \times C}$. Then, $H_j$ is evenly split along the channel dimension into two parts, which are processed by different branches for feature extraction. The first branch consists of a

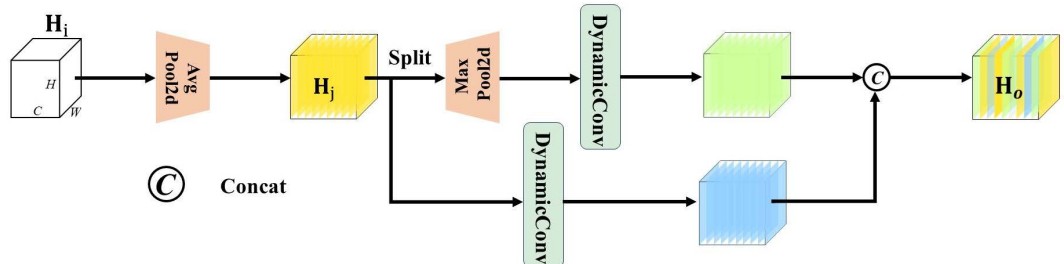

**Fig 5. Structure of the Dynamic Downsampling module.**

3×3 max pooling layer followed by a 1×1 dynamic convolution layer. The second branch is composed of a 3×3 dynamic convolution layer. At this stage, the feature maps in both branches have a size of $H \times W \times C/8$. Finally, the feature maps from both branches are concatenated to obtain the output feature map of DyDown, denoted as $H_o \in \mathbb{R}^{H \times W \times C/4}$.

## 4. Experiments and analysis

This section first introduces the four datasets: Insulator-DET, CPLID, IDID, and VOC07 + 12. Detailed information is presented in Table 1. Next, the experimental metrics and environment are explained. Subsequently, by comparing the performance of TLINet with the YOLOv9 and YOLOv10 series on the Insulator-DET dataset, the superiority of TLINet is demonstrated. Additionally, TLINet's detection results are compared with other state-of-the-art methods on the Insulator-DET, CPLID, IDID, and VOC07 + 12 datasets, fully verifying its feasibility and generalization capability. Furthermore, ablation experiments are conducted to validate the effectiveness of the proposed improvements, and the generalization of the proposed MBPTB, CGFFN, and DyDown structures is verified using the YOLOv5 framework. Finally, the backbone network, feature fusion network, and downsampling module of the baseline model are individually replaced and compared with TLINet. The experimental results further confirm the superiority of TLINet.

### 4.1. Dataset Description

To evaluate the effectiveness and generalization capability of the proposed method in insulator defect detection, four datasets were selected for the experiments in this study. The detailed information on these datasets is provided below.

(1) Insulator-DET: The Insulator-DET dataset was constructed by our team using insulator defect images collected by UAV. This dataset includes nine types of defects: glass-dirty, glass-loss, polymer, polymer-dirty, two-glass, broken-disc, insulator, flashover, and snow. The Insulator-DET dataset contains 2,150 images, which are divided into training, validation, and test sets in a ratio of 8:1:1. Each image has a resolution of 640×640. The Insulator-DET dataset provides a rich and well-annotated set of samples for insulator defect detection, supporting model training and

**Table 1. Details of the five public datasets.**

| Dataset | Class type | Total images | Images size | Num of train set | Num of val set | Num of test set |
|---|---|---|---|---|---|---|
| Insulator-DET | 9 | 2150 | 640×640 | 1720 | 215 | 215 |
| CPLID-D | 1 | 2220 | 1152×864 | 1776 | 444 | 100 |
| CPLID-N | 1 | 2520 | 1152×864 | 2016 | 504 | 240 |
| IDID | 2 | 3031 | 640×640 | 2424 | 303 | 304 |
| VOC07 + 12 | 20 | 22136 | 550×375 | 15495 | 4427 | 2214 |

validation while advancing the development of automated insulator defect detection technology. The dataset is available at https://www.kaggle.com/datasets/mazilishanglx/insulator-det.

(2) CPLID: The China Power Line Insulator Dataset (CPLID) is an open-source image dataset specifically designed for defect detection of power line insulators. It consists of two parts: 600 images of normal insulators and 248 synthesized images of defective insulators. The normal insulator images provide visual references of insulators in proper working conditions, serving as a baseline for defect detection. To simulate potential issues during UAV image transmission in real-world scenarios, we divided CPLID into two subsets: CPLID-D and CPLID-N. From these, we selected 100 and 240 images respectively as their corresponding test sets. Due to the limited number of defective insulator samples in real cases, data augmentation was used to synthesize insulator images. Further data augmentation strategies—including cropping, translation, brightness adjustment, noise addition, rotation, mirroring, and cutout [45]—were applied to both subsets. As a result, CPLID-D and CPLID-N were expanded to 2,220 and 2,520 images respectively. During network training, each dataset was randomly split into training and validation sets at a ratio of 8:2. The dataset is available at https://www.kaggle.com/datasets/mazilishanglx/cplid-dcplid-n.

(3) IDID: This is a publicly available insulator defect dataset provided by the IEEE competition. The Insulator Defect Image Dataset (IDID) consists of high-quality, annotated images of transmission line insulators. These images primarily contain two types of defects: flashover and broken. The IDID dataset contains 3,031 images, which are divided into training, validation, and test sets in a ratio of 8:1:1. Each image has a resolution of 640 × 640. The dataset is available at https://www.kaggle.com/datasets/mazilishanglx/idid-dataset.

(4) VOC07 + 12: In this study, the VOC2007 and VOC2012 datasets were combined and named the VOC07 + 12 dataset. VOC2007 covers 20 object categories and is widely used for object detection tasks. VOC2012 extends this dataset by incorporating additional image segmentation annotations. By merging these two datasets, the model's ability to adapt to various object types, complex backgrounds, and environmental variations is enhanced. The combined dataset provides a broader range of testing scenarios, facilitating a comprehensive evaluation of the detection model's performance under diverse conditions. The VOC07 + 12 dataset contains 22,136 images, which are divided into training, validation, and test sets in a ratio of 7:2:1. Each image has a resolution of 550 × 375. The dataset is available at https://www.kaggle.com/datasets/mazilishanglx/voc07-12.

## 4.2. Training Strategies and Implementation Details

To ensure the reproducibility of the experimental results, all experiments were conducted on the same high-performance deep learning server. The server configuration is as follows: an Intel i9-13900K 3.00GHz CPU, an NVIDIA GeForce RTX 3090−24GB GPU, and the Windows 10 operating system. Python 3.10.14, CUDA 11.8, and PyTorch 2.2.2 were used as the deep learning framework. The hyperparameter settings for model training are shown in Table 2, with the input image size set to 640 × 640.

For hyperparameter selection, the initial learning rate was set to 1e-3, which was experimentally verified to ensure stable convergence while avoiding instability or slow training due to excessively large or small learning rates. The batch size was set to 16, striking a balance between memory consumption and training stability while effectively utilizing GPU computational power. The momentum was set to 0.937, which helps accelerate the Adam optimization algorithm, reducing oscillations during training and ensuring smooth convergence. The weight decay was set to 5e-4 to control model complexity and prevent overfitting, thereby enhancing generalization performance. A cosine learning rate decay strategy was employed to gradually decrease the learning rate in later training stages, helping the model avoid oscillations and improve final accuracy. The total number of training epochs was set to 200 to ensure the model achieves optimal performance after sufficient learning. Finally, Mosaic augmentation was enabled throughout the training process. This augmentation

**Table 2. Initialization Parameters Of Our Method.**

| Parameters | Value | Note |
|---|---|---|
| Decay strategy | cosine | Cosine Annealing Function |
| Learning rate | 1e-3 | Initial learning rate |
| Optimizer | Adam | Fast convergence |
| Momentum | 0.937 | Improve optimization stability |
| Total epochs | 200 | Training cycles for convergence |
| Weight decay | 5e-4 | Prevent overfitting |
| Close mosaic | 0 | Disable augmentation |
| Batch size | 16 | Number of samples per iteration |

technique enhances the model's ability to adapt to various scenarios. In real-world applications, objects may be partially occluded, blurred, or affected by background noise, making them less distinguishable. Continuous training with Mosaic augmentation improves the model's robustness and accuracy, particularly in complex backgrounds or low-quality images.

### 4.3. Evaluation metrics

To comprehensively evaluate the accuracy and real-time performance of the model, the following key performance metrics are adopted: Precision(P), Recall(R), Average Precision (AP), Mean Average Precision (mAP), F1 Score, and Frames Per Second (FPS). These metrics assess the overall performance of the detector in object detection tasks, including detection accuracy, recognition capability, overall detection performance, and inference speed.

1) Precision and Recall: Precision represents the proportion of predicted positive samples that are actually positive, measuring the accuracy of the detector in positive class detection. Recall indicates the proportion of actual positive samples correctly predicted by the model, reflecting the detector's ability to identify targets. The definitions of precision and recall are as follows:

$$Precision = \frac{TP}{TP + FP}$$

$$(3)$$

$$Recall = \frac{TP}{TP + FN}$$

$$(4)$$

Where TP (True Positive) represents correctly detected targets, FP (False Positive) denotes background regions mistakenly identified as targets, and FN (False Negative) refers to targets that the detector failed to identify, incorrectly treating them as background.

2) Average Precision: In object detection tasks, AP is a key metric for evaluating model performance. It quantifies the balance between precision and recall for a specific class by computing the area under the precision-recall curve. The formula for AP is defined as follows:

$$AP_n = \int_0^1 P(r)dr$$

$$(5)$$

Where n represents the number of classes, P(r) denotes the precision-recall curve, and the computed integral represents the AP value for the corresponding class.

3) Mean Average Precision: To comprehensively assess the model's performance across all classes, the mAP is utilized. It represents the average of the AP values for each class in the dataset, providing an overall evaluation of the

detector's effectiveness. Specifically, mAP50 refers to the mAP value calculated when the Intersection over Union (IoU) threshold between the predicted and ground truth bounding boxes is set to 0.5. The formula for calculating mAP is as follows:

$$mAP = \frac{1}{N}\sum_{n=1}^{N}AP_n$$

(6)

Where N is the total number of classes, and AP_n is the Average Precision for class n.

4) F1 Score: To balance both precision and recall, the F1 score is introduced as the harmonic mean of these two metrics. The F1 score provides a more comprehensive assessment of the detection model's performance. The formula for calculating the F1 score is:

$$F1\ score = 2\times \frac{Precision \times Recall}{Precision + Recall}$$

(7)

5) FPS: To evaluate the real-time performance of the detector, FPS is used as a key metric. FPS indicates the number of images the detector can process per second, providing an assessment of its inference speed. A higher FPS value signifies improved real-time performance of the detector.

$$FPS = \frac{1}{T_{per}}$$

(8)

where $T_{per}$ denotes the time taken by the detector to process a single defect image.

### 4.4. Results on insulator-DET, CPLID and IDID dataset

**Comparison of TLINet with YOLOv8, YOLOv9, and YOLOv10 Series** In comparison with the YOLOv8, YOLOv9, and YOLOv10 series, TLINet demonstrates significant advantages, as shown in Tables 3–5. Specifically, TLINet's parameter count is only 9.6% of YOLOv8m, its computational complexity is 8.6% of YOLOv8m, and its inference speed is 1.5 times of YOLOv8m. This makes TLINet highly advantageous for deployment on resource-constrained UAV devices.

**Table 3. Performance Comparison of Tlinet and Yolov8 Families.**

| Methods | Params↓ | GFLOPS↓ | FPS↑ | mAP$_{50}$↑ | F1 scores↑ | P↑ | R↑ |
|---|---|---|---|---|---|---|---|
| YOLOv8n | 3.0 | 8.1 | **308** | 51 | 0.51 | **0.728** | 0.499 |
| YOLOv8s | 11.1 | 28.5 | 274 | 55 | **0.56** | 0.578 | 0.555 |
| YOLOv8m | 25.8 | 78.7 | 135 | 56.3 | **0.56** | 0.518 | 0.596 |
| YOLOv8l | 43.6 | 164.9 | 85 | **58.8** | **0.56** | 0.657 | **0.597** |
| TLINet | **2.48** | **6.8** | 204 | 56.3 | 0.55 | 0.641 | 0.588 |

**Table 4. Performance Comparison Of Tlinet And Yolov9 Families.**

| Methods | Params↓ | GFLOPS↓ | FPS↑ | mAP$_{50}$↑ | F1 scores↑ | P↑ | R↑ |
|---|---|---|---|---|---|---|---|
| YOLOv9t | **1.97** | 7.6 | 151 | 53.1 | 0.51 | **0.713** | 0.507 |
| YOLOv9s | 7.17 | 26.7 | 106 | **62** | 0.59 | 0.577 | 0.616 |
| YOLOv9m | 20.16 | 77 | 83 | 60.1 | **0.61** | 0.607 | **0.62** |
| YOLOv9c | 25.2 | 102.4 | 80 | 57 | 0.57 | 0.607 | 0.56 |
| TLINet | 2.48 | **6.8** | **204** | 56.3 | 0.55 | 0.641 | 0.588 |

**Table 5. Performance Comparison Of Tlinet And Yolov10 Families.**

| Methods | Params↓ | GFLOPS↓ | FPS↑ | $mAP_{50}$↑ | F1 scores↑ | P↑ | R↑ |
|---|---|---|---|---|---|---|---|
| YOLOv10n | **2.26** | **6.5** | **270** | 45.7 | 0.45 | 0.551 | 0.438 |
| YOLOv10s | 7.22 | 21.4 | 269 | 55.1 | 0.52 | **0.662** | 0.514 |
| YOLOv10m | 15.31 | 58.9 | 120 | 51.7 | 0.49 | 0.473 | 0.51 |
| YOLOv10l | 24.31 | 120 | 78 | 49.9 | 0.49 | 0.621 | 0.473 |
| TLINet | 2.48 | 6.8 | 204 | **56.3** | **0.55** | 0.641 | **0.588** |

Furthermore, in comparison with the YOLOv9 and YOLOv10 series, TLINet exhibits similar advantages. TLINet achieves an FPS of 204 and an mAP50 of 56.3%, which are significantly higher than those of YOLOv9t and YOLOv10m. Finally, TLINet's recall and F1 score are also significantly higher than those of YOLOv8n, indicating that TLINet not only effectively balances precision and recall but also substantially reduces the missed detection. However, TLINet's mAP50 is significantly lower than that of larger models such as YOLOv8l and YOLOv9c, indicating that there is still room for improvement in tasks requiring higher accuracy. Fig 6 illustrates the performance of each method.

**Comparison with state-of-the-art methods:** To further demonstrate the superiority of TLINet, it was compared with several state-of-the-art methods on the Insulator-DET, IDID, and CPLID benchmark datasets. The models included in the comparison are: Faster R-CNN, CenterNet, EfficientDet-D1, RT-DETR, FCOS, YOLOX, YOLOv3-tiny, YOLOv5n, YOLOv6n, YOLOv7-tiny, YOLOv8n, YOLOv9t, YOLOv10n, YOLOv11n, YOLOv12n, Hyper-YOLO, and GELAN-t. As a classic two-stage object detection model, Faster R-CNN serves as a valuable reference for detection tasks involving complex scenes. Centernet localizes objects by predicting their center points, which enables it to achieve high detection accuracy. EfficientDet-d1 is a lightweight and efficient object detection model that offers high computational efficiency with low resource consumption. As an anchor-free detection method, FCOS achieves high detection accuracy with low computational resources. The YOLO model's high accuracy and real-time performance provide significant advantages in object detection tasks, making it particularly suitable for UAV platforms with limited resources. Compared to other YOLO versions, YOLOX is optimized for both accuracy and efficiency, making it particularly advantageous in defect detection tasks of various scales. GELAN-t offers excellent performance and efficiency. Hyper-YOLO is an optimized

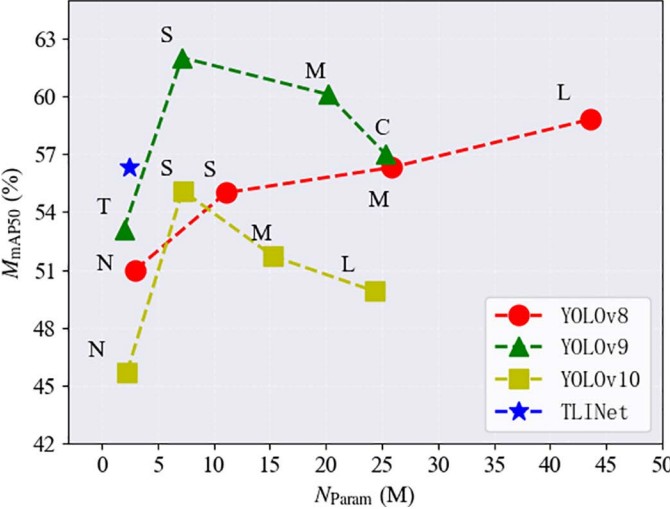

**Fig 6. Performance of TLINet, YOLOv8, YOLOv9 and YOLOv10 on Insulator-DET dataset.**

**TablE 6. Performance Comparison Of Each Method On Insulator-Det Datasets.**

| Method | P↑ | R↑ | mAP$_{50}$↑ | FPS↑ | Params↓ |
|---|---|---|---|---|---|
| Faster-rcnn | 0.230 | 0.200 | 23.76 | 18 | 136.85 |
| Centernet | 0.667 | 0.172 | 39.5 | 65 | 32.66 |
| EfficientDet-d1 | 0.420 | 0.170 | 26.82 | 23 | 3.83 |
| RT-DETR | 0.410 | 0.434 | 42.4 | 87 | 8.33 |
| FCOS | 0.554 | 0.467 | 57.2 | 56 | 32.13 |
| YOLOX | 0.572 | 0.557 | **62.0** | 110 | 54.21 |
| YOLOv3-tiny | 0.577 | 0.511 | 50.1 | **394** | 12.13 |
| YOLOv5n | 0.592 | 0.541 | 49.8 | 292 | **1.76** |
| YOLOv6n | 0.533 | 0.491 | 46.8 | 306 | 4.23 |
| YOLOv8n | **0.726** | 0.499 | 51.0 | 308 | 3.00 |
| YOLOv9t | 0.713 | 0.507 | 53.1 | 151 | 1.97 |
| YOLOv10n | 0.551 | 0.438 | 45.7 | 270 | 2.26 |
| YOLOv11n | 0.526 | 0.509 | 52.0 | 260 | 2.58 |
| YOLOv12n | 0.400 | 0.528 | 46.8 | 160 | 2.55 |
| TLINet | 0.641 | **0.588** | 56.3 | 204 | 2.48 |

**Table 7. Performance Comparison Of Each Method On Cplid-D And Cplid-N Datasets.**

| Method | CPLID-D | | | CPLID-N | | |
|---|---|---|---|---|---|---|
| | mAP$_{50-95}$↑ | FPS↑ | Params↓ | mAP$_{50-95}$↑ | FPS↑ | Params↓ |
| YOLOv3-tiny | 81.9 | **401** | 12.12 | 62.7 | **394** | 12.13 |
| YOLOv5n | 81.6 | 292 | **1.76** | 69.7 | 296 | **1.76** |
| YOLOv6n | 80.7 | 300 | 4.23 | 69 | 310 | 4.23 |
| YOLOv8n | 82 | 307 | 3.00 | 70.5 | 306 | 3.00 |
| YOLOv9t | 80.9 | 149 | 1.97 | 71.1 | 150 | 1.97 |
| YOLOv10n | 80.1 | 274 | 2.26 | 65.2 | 275 | 2.26 |
| TLINet | **89.9** | 205 | 2.48 | **82.6** | 206 | 2.48 |

**Table 8. Performance Comparison Of Each Method IDID Datasets.**

| Method | P↑ | R↑ | mAP$_{50}$↑ | FPS↑ | Params↓ |
|---|---|---|---|---|---|
| YOLOv7-tiny | 0.935 | 0.954 | 96.8 | **337** | 6.01 |
| Hyper-YOLO | 0.941 | **0.958** | **97.6** | 199 | 3.94 |
| GELAN-t | 0.913 | 0.920 | 95.4 | 135 | **1.87** |
| YOLOv10n | 0.908 | 0.909 | 95.8 | 228 | 2.69 |
| YOLOv11n | **0.953** | 0.938 | 97.4 | 259 | 2.58 |
| YOLOv12n | 0.950 | 0.952 | **97.6** | 177 | 2.55 |
| RT-DETR | 0.929 | 0.931 | 95.1 | 82 | 8.28 |
| TLINet | 0.950 | 0.944 | **97.6** | 204 | 2.48 |

YOLO variant designed for specific applications, achieving higher detection accuracy and faster inference speed. As a Transformer-based object detection model, RT-DETR provides real-time detection capabilities. By introducing RT-DETR, the performance differences between the Transformer architecture and the YOLO architecture in UAV-based object detection tasks can be analyzed. Table 6–8 record the performance of different methods on Insulator-DET, CPLID and IDID datasets respectively.

**Insulator-DET:** TLINet exhibits excellent performance in terms of recall and mAP50. Specifically, TLINet achieves the highest recall, while its mAP50 is only lower than that of YOLOX and FCOS. In terms of inference speed, although TLINet ranks seventh, its FPS is still significantly higher than that of YOLOX and FCOS. TLINet sacrifices some inference speed in exchange for a significant improvement in accuracy, making it a viable solution for real-time detection tasks that require high precision. In terms of the number of parameters, although TLINet is higher than YOLOv5n, YOLOv9t, and YOLOv10n, its mAP50 and FPS are significantly better than those of YOLOv9t. Compared to YOLOv10n, TLINet's FPS decreased by one-third, but its mAP50 increased by 11.6%. The trade-off between the number of parameters, FPS, and mAP50 allows TLINet to still run efficiently on UAV platforms with limited computational resources.

The structural design of TLINet also highlights its advantages. The Transformer component in MBPTB and the dual-path structure in DyDown significantly enhance the response capability to multi-scale defects. Compared to traditional two-stage models like Faster R-CNN, TLINet has faster inference speed. Although the accuracy of FCOS is similar to TLINet, its parameter count is higher, and its FPS is much lower than that of TLINet. Therefore, TLINet performs better on devices with limited computational resources. In terms of accuracy and inference speed, TLINet significantly outperforms RT-DETR. Analysis shows that RT-DETR uses Transformer as its encoder, which greatly increases its computational burden. In contrast, MBPTB in TLINet significantly reduces the computational overhead caused by the Transformer. However, compared to baseline model, TLINet's inference speed has decreased, which may lead to performance degradation in multi-object scenarios. Analysis indicates that the dual-path structure in DyDown requires more feature fusion and computation, which may increase the inference time. Additionally, the self-attention mechanism in MBPTB adds to the computational load, resulting in slower inference speed. To address this issue, MBPTB can adopt more efficient and lightweight Transformer variants to reduce computational complexity. Furthermore, the DyDown module can be replaced with a single-path downsampling module to reduce unnecessary path computation and feature fusion. Through these optimizations, TLINet's inference speed can be effectively improved without significantly affecting accuracy. To provide a clearer illustration of the quantitative analysis results of each method on the Insulator-DET dataset, this study uses scatter plots for visual representation, as shown in Fig 7. The detection results are shown in Fig 8.

**CPLID:** The validation results on the CPLID-D and CPLID-N datasets are similar to those on the Insulator-DET dataset, with TLINet achieving optimal performance in mAP50-95. Specifically, compared to the baseline model, TLINet's

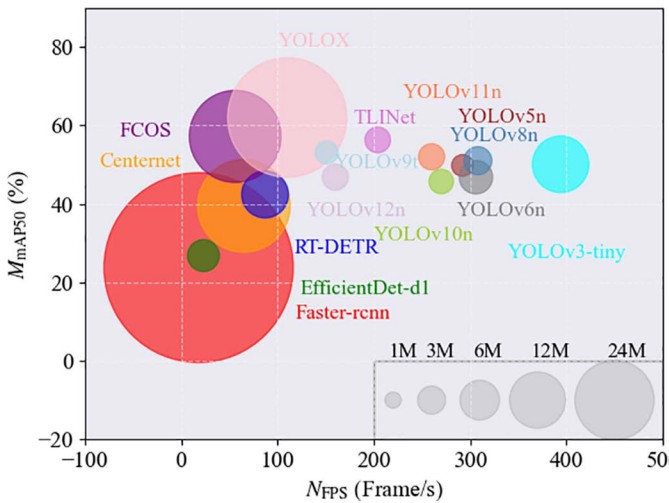

**Fig 7. Experimental results for each method on the Insulator-DET dataset.**

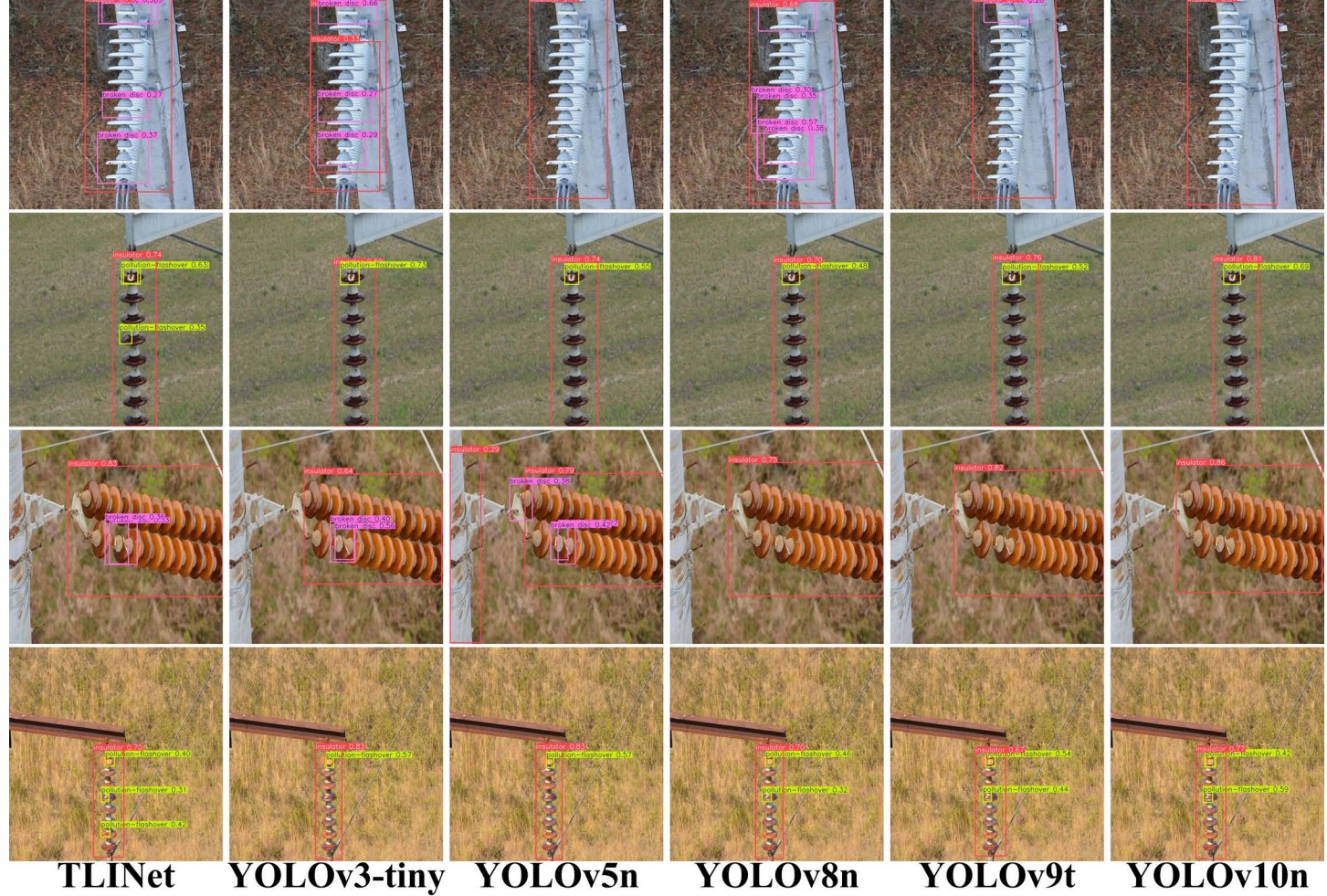

**Fig 8. Some detection results of different models on the Insulator-DET dataset.**

mAP50-95 improved by 7.9% and 12.1%, respectively. This further demonstrates TLINet's strong capability in recognizing and locating insulator defects. The detection results are shown in Fig 9.

**IDID:** In terms of mAP50, TLINet matches YOLOv12n and Hyper-YOLO, all achieving optimal performance. However, TLINet has a parameter count of 2.48M, which is lower than both YOLOv12n and Hyper-YOLO. Additionally, TLINet's FPS is 204, which is higher than YOLOv12n and Hyper-YOLO. This indicates that, compared to the current state-of-the-art lightweight models, TLINet not only maintains detection accuracy but also extends its advantages in terms of parameter count and inference speed. Therefore, compared to YOLOv12n and Hyper-YOLO, TLINet is better suited for deployment on resource-constrained drone platforms. In terms of inference speed, YOLOv7-tiny achieves the best performance. However, compared to TLINet, YOLOv7-tiny's precision and mAP50 decrease by 1.5% and 0.8%, respectively. Furthermore, YOLOv7-tiny has a parameter count of 6.01M, which is 2.4 times higher than TLINet. As a result, YOLOv7-tiny's deployment on UAV platforms is still significantly limited. The detection results are shown in Fig 10.

Overall, TLINet strikes a good balance between accuracy and inference speed. It improves the detection capability of multi-scale defects through innovative design. While it may not perform as well as some YOLO models in tasks with extremely high real-time requirements, TLINet still offers clear advantages in terms of computational resource consumption and detection accuracy, making it particularly well-suited for deployment in resource-constrained environments.

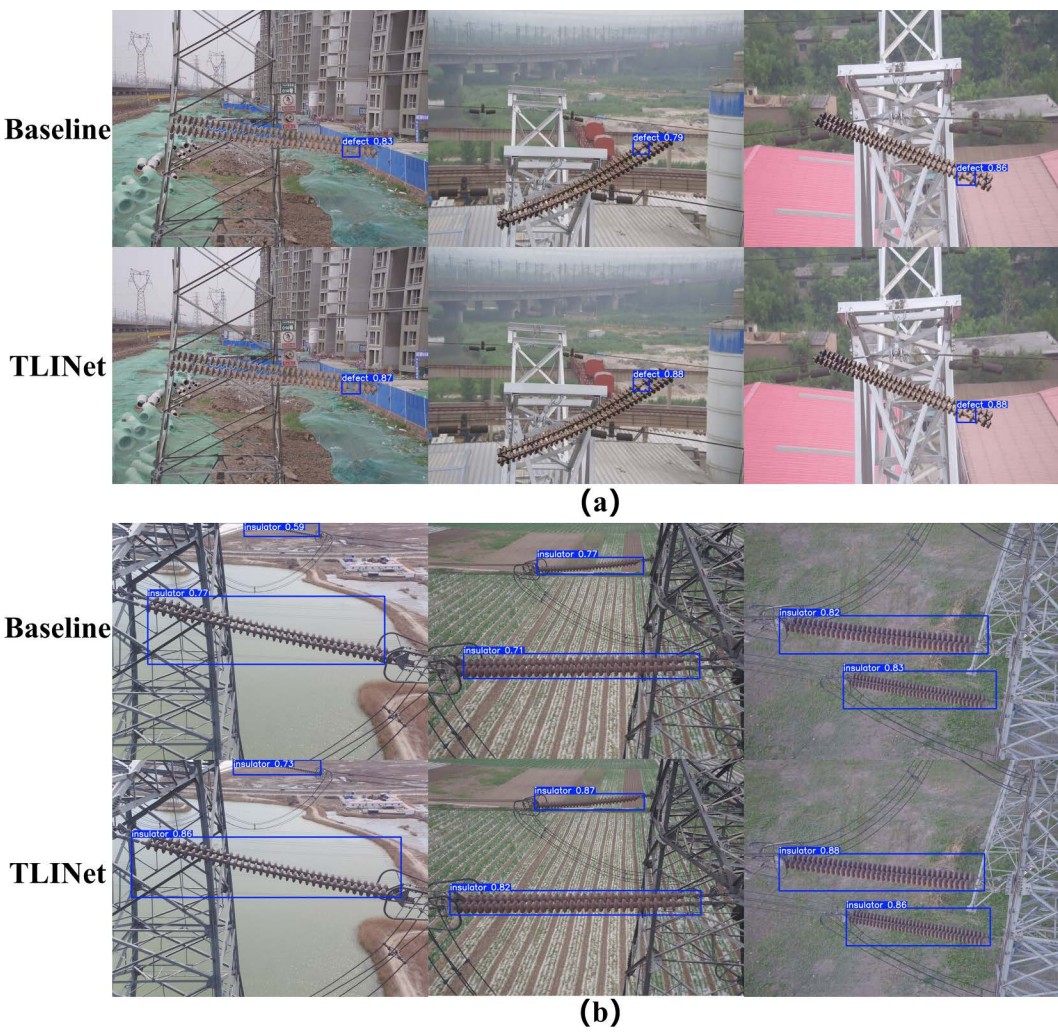

**Fig 9. Some detection results of the baseline model and TLINet on the CPLID dataset. (a) CPLID-D (b) CPLID-N.**

## 4.5. Generalization experiment

To evaluate the robustness and generalization performance of TLINet, experiments were conducted on the VOC07 + 12 dataset. Table 9 presents the results of each method on the VOC07 + 12 dataset.

According to the table, TLINet performs well in terms of precision and mAP50. Specifically, compared to YOLOv8n, TLINet improves mAP50 and precision by 0.4% and 1.3%, respectively, while reducing the parameter count by 0.52M. Although YOLOv7-tiny achieves a 7.2% higher mAP50 than TLINet, TLINet's parameter count is only 41% of that of YOLOv7-tiny. Compared to YOLOv12n, TLINet's mAP50 is 0.9% lower, but its FPS increases by 30. Additionally, TLINet has a slightly lower parameter count than YOLOv12n. This lightweight design enables TLINet to maintain high accuracy while reducing computational burden, making it suitable for deployment on resource-limited UAV platforms.

Compared to YOLOv8n, TLINet achieves a balance between model size and inference speed. While it sacrifices some inference speed, the improvements in precision and mAP50 make it a highly promising object detection model. TLINet also demonstrates strong generalization capabilities. The experimental results on the VOC07 + 12 dataset indicate that TLINet can consistently enhance detection performance across different datasets and complex scenarios. However,

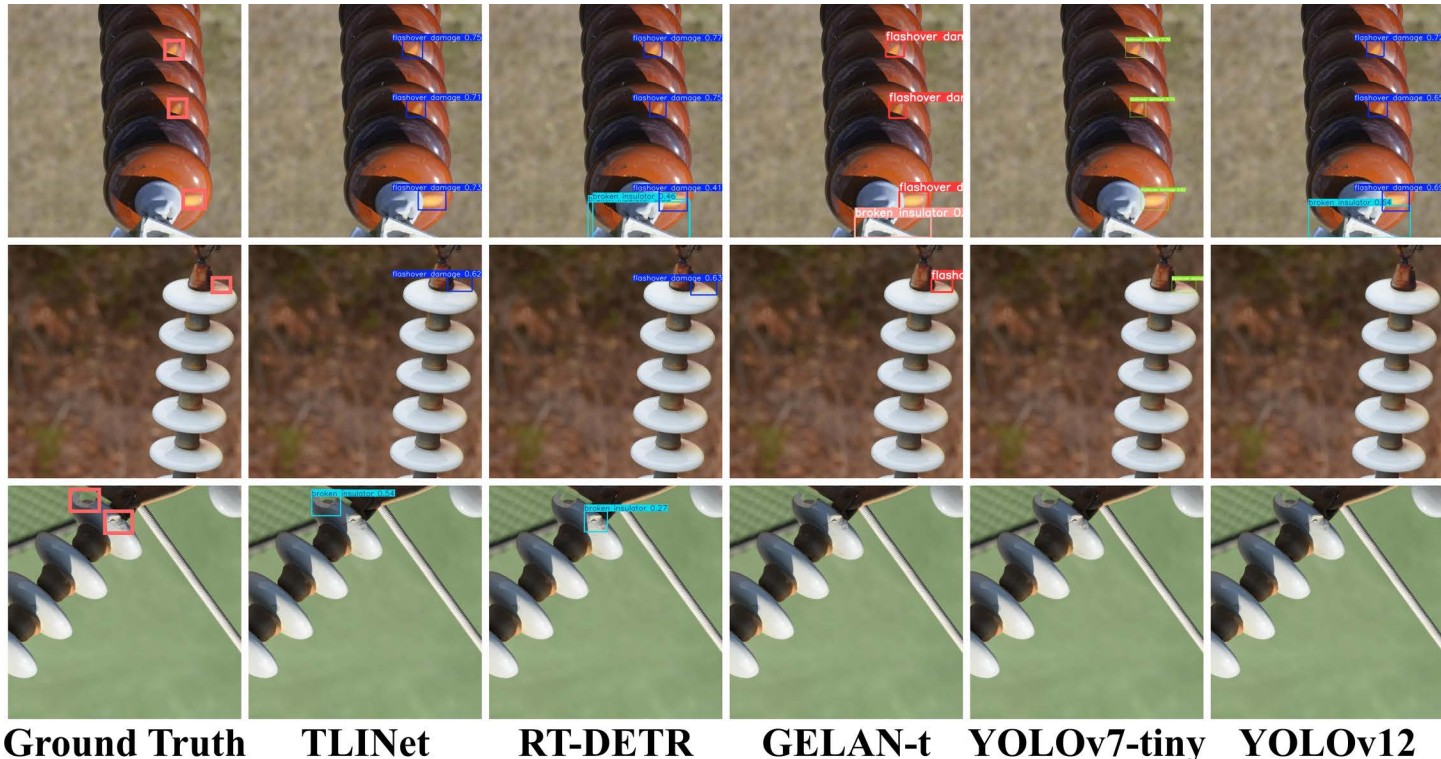

| Ground Truth | TLINet | RT-DETR | GELAN-t | YOLOv7-tiny | YOLOv12 |

**Fig 10. Some detection results of different models on IDID dataset.**

**Table 9. Performance comparison of each method on voc07 + 12 datasets.**

| Method | P↑ | R↑ | mAP$_{50}$↑ | FPS↑ | Params↓ |
|---|---|---|---|---|---|
| YOLOv5n | 0.389 | 0.401 | 32.1 | **356** | **1.77** |
| YOLOv7-tiny | **0.517** | **0.506** | **47.8** | 329 | 6.05 |
| YOLOv8n | 0.420 | 0.473 | 40.2 | 317 | 3.00 |
| GELAN-t | 0.414 | 0.480 | 39.9 | 131 | 1.91 |
| YOLOv10n | 0.431 | 0.472 | 39.8 | 235 | 2.70 |
| YOLOv11n | 0.448 | 0.475 | 41.7 | 263 | 2.58 |
| YOLOv12n | 0.435 | 0.488 | 41.5 | 175 | 2.56 |
| TLINet | 0.433 | 0.473 | 40.6 | 205 | 2.48 |

TLINet's main limitation is the decrease in inference speed. Although its inference speed is higher than that of YOLOv12n and GELAN-t, the incorporation of dynamic convolution and Transformer blocks increases computational complexity, leading to slower inference. Compared to YOLOv7-tiny and YOLOv5n, TLINet's inference speed is lower, indicating that it may not be suitable for applications with extremely high real-time requirements.

In conclusion, TLINet excels in balancing model size and detection accuracy. Although its inference speed is slightly reduced, further optimization of the model structure or adoption of more efficient computational methods could help TLINet achieve a better balance between accuracy and efficiency, enhancing its applicability in real-time demanding tasks.

## 4.6. Ablation analysis

To evaluate the stability and reliability of the model, multiple independent experiments were conducted on the Insulator-DET validation set using the baseline model combined with different improvement strategies. The mean and standard deviation were calculated, and the results show that the performance fluctuations across various metrics are minimal, indicating good stability. Detailed statistical results are presented in Table 10. Each improvement component contributed to an increase in detection accuracy to varying degrees.

Specifically, after introducing CGFFN, TLINet's mAP50 increased by 1.9%, while the parameter count decreased by 0.37M. With the addition of DyDown, mAP50 improved by 4.7% without affecting the parameter count. Incorporating MBPTB further enhanced TLINet's recall and mAP50 by 3.1% and 3.2%, respectively. Ultimately, compared to the baseline method, TLINet's recall increased by 8.9%, mAP50 improved by 5.3%, and the parameter count was reduced by 0.52M. However, despite the significant improvements in detection accuracy, the complex structures of these components led to a decrease in FPS. In the following sections, this study will conduct a detailed experimental analysis of each improvement strategy in TLINet, including CGFFN, DyDown, and MBPTB.

**1) Effectiveness of CGFFN:** As shown in Table 11, CGFFN demonstrates strong advantages across multiple categories, particularly in mAP50. For example, in the glass-loss category, mAP50 increased from 38.3% to 56.6%, an improvement of 18.3%. The enhancement indicates that CGFFN effectively improves the detector's accuracy in distinguishing challenging defect types, thereby reducing false detections caused by background noise. Similarly, in the polymer category, mAP50 increased from 26.3% to 42.7%, a gain of 16.4%, further demonstrating CGFFN's capability in feature fusion when handling low-contrast defects. By introducing MBCGM, CGFFN not only strengthens the model's ability to learn from complex backgrounds but also enhances its recognition of blurry or low-contrast defects. In terms of recall, CGFFN also performs significantly well, particularly in the glass-loss category, where recall increased from 0.417 to 0.583, a gain of 0.166. This indicates that CGFFN enhances the model's ability to detect fine-grained defects. Other categories, such as polymer and flashover, also saw recall improvements, increasing from 0.375 to 0.375 and from 0.657 to 0.575, respectively, demonstrating CGFFN's adaptability and effectiveness across different defect types. By optimizing feature fusion, CGFFN helps the detector better recognize targets, especially in cases of strong background noise and blurry objects. Despite its strong performance in most categories, CGFFN still shows some limitations in certain defect types. For instance, in the glass-dirty category, recall decreased from 0.510 to 0.475, and mAP50 dropped by 3.7%. This suggests that when dealing with relatively simple defects, CGFFN's excessive background optimization may negatively impact the model's ability to recognize the defect itself. In the polymer-dirty category, recall increased from 0.065 to 0.169, and mAP50 improved from 33.6% to 37.7%, but precision dropped significantly from 1.000 to 0.648. Although the increase in recall helps detect more targets, excessive background optimization may have led to a higher false detection rate. In the two-glasses category, mAP50 decreased slightly from 99.5% to 97.9%, and recall dropped from 1.000 to 0.905, revealing CGFFN's limitations in handling highly distinguishable defects.

**Table 10. Performance Comparison of The Baseline Combination with Different Strategies.**

| Method | CGFFN | DyDown | MBPTB | Params↓ | mAP$_{50}$↑ | FPS↑ | P↑ | R↑ |
|---|---|---|---|---|---|---|---|---|
| Baseline | – | – | – | 3.0 | 51.0±0.23 | **308±2.42** | **0.726±0.007** | 0.499±0.009 |
| | √ | – | – | 2.63 | 52.9±0.25 | 281±2.61 | 0.658±0.005 | 0.504±0.004 |
| | – | √ | – | 3.0 | 55.7±0.19 | 190±1.72 | 0.637±0.006 | 0.578±0.005 |
| | – | – | √ | 2.89 | 54.2±0.17 | 255±1.58 | 0.704±0.002 | 0.530±0.004 |
| | √ | √ | – | 2.59 | 55.0±0.18 | 252±1.58 | 0.687±0.003 | 0.529±0.003 |
| | √ | √ | √ | **2.48** | **56.3±0.19** | 204±1.14 | 0.641±0.002 | **0.588±0.002** |

**Table 11. The Impact Of Different Enhancement Strategies On Different Types Of Defects.**

| Methods | Defect type | P↑ | R↑ | mAP$_{50}$↑ |
|---|---|---|---|---|
| Baseline | glass-dirty | 0.460 | 0.510 | 47.1 |
| | glass-loss | 0.589 | 0.417 | 38.3 |
| | polymer | 0.422 | 0.375 | 26.3 |
| | polymer-dirty | 1.000 | 0.065 | 33.6 |
| | two-glasses | 0.968 | 1.000 | 99.5 |
| | broken disc | 0.591 | 0.589 | 60.7 |
| | insulator | 0.850 | 0.879 | 90.8 |
| | flashover | 0.658 | 0.657 | 61.7 |
| | snow | 1.000 | 0.000 | 0.00 |
| Baseline+CGFFN | glass-dirty | 0.538 | 0.475 | 43.4(3.7↓) |
| | glass-loss | 0.541 | 0.583 | 56.6(18.3↑) |
| | polymer | 0.271 | 0.375 | 42.7(16.4↑) |
| | polymer-dirty | 0.648 | 0.169 | 37.7(4.1↑) |
| | two-glasses | 0.849 | 0.905 | 97.9(1.6↓) |
| | broken disc | 0.563 | 0.547 | 53.2(7.5↓) |
| | insulator | 0.855 | 0.904 | 91.1(0.3↑) |
| | flashover | 0.665 | 0.575 | 54.8(6.9↓) |
| | snow | 1.000 | 0.000 | 0.00(-) |
| Baseline+MBPTB | glass-dirty | 0.472 | 0.475 | 44.4(2.7↓) |
| | glass-loss | 0.646 | 0.500 | 50.6(12.3↑) |
| | polymer | 0.568 | 0.494 | 34.3(8↑) |
| | polymer-dirty | 0.641 | 0.164 | 45.1(11.5↑) |
| | two-glasses | 0.859 | 1.000 | 98.9(0.6↓) |
| | broken disc | 0.646 | 0.611 | 61.4(0.7↑) |
| | insulator | 0.864 | 0.911 | 92.3(1.5↑) |
| | flashover | 0.644 | 0.617 | 59.8(1.9↓) |
| | snow | 1.000 | 0.000 | 0.00(-) |
| Baseline+DyDown | glass-dirty | 0.458 | 0.675 | 55.5(8.4↑) |
| | glass-loss | 0.712 | 0.621 | 59.4(21.1↑) |
| | polymer | 0.295 | 0.375 | 32.1(5.8↑) |
| | polymer-dirty | 0.409 | 0.253 | 35.1(1.8↑) |
| | two-glasses | 0.846 | 1.000 | 99.5(-) |
| | broken disc | 0.508 | 0.695 | 64.0(3.3↑) |
| | insulator | 0.875 | 0.917 | 92.7(1.9↑) |
| | flashover | 0.628 | 0.671 | 62.4(0.7↑) |
| | snow | 1.000 | 0.000 | 0.00(-) |

In addition, this study compares the impact of various common feature fusion networks on performance to verify the effectiveness of CGFFN. As shown in Table 12, CGFFN significantly improves both recall and mAP50 compared to the original FPN, with increases of 0.5% and 1.9%, respectively. Compared to advanced architectures such as BiFPN and MAFPN [46], CGFFN also demonstrates clear advantages. Fig 11 visualizes the heatmaps of different feature fusion networks. The heatmap generated by CGFFN exhibits distinct red-highlighted areas along the defect boundaries, which closely align with the actual defect locations. In contrast, the highlighted regions in other methods often appear blurred

**Table 12. Compare the Performance of Different Feature-Fusion Network Architectures.**

| Methods | P↑ | R↑ | mAP$_{50}$↑ | Params↓ | FPS↑ |
|---|---|---|---|---|---|
| Baseline | **0.726** | 0.499 | 51.0 | 3.00 | **308** |
| Baseline with BiFPN | 0.573 | **0.572** | 51.2 | **1.99** | 265 |
| Baseline with MAFPN | 0.597 | 0.518 | 52.2 | 2.98 | 264 |
| Baseline with CGFFN | 0.658 | 0.504 | **52.9** | 2.63 | 281 |

or diffused at the defect edges, leading to false detections or missed detections. Furthermore, CGFFN's heatmap shows almost no significant intensity distribution in defect-free background regions, maintaining a uniform color, whereas other methods' heatmaps display noticeable high-intensity regions in the background, likely due to noise or irrelevant background interference.

The above analysis confirms that CGFFN is well-suited for complex insulator defect detection tasks and holds significant practical value and application potential.

**2) Effectiveness of DyDown:** As shown in Table 11, the DyDown demonstrated significant advantages in detecting glass-loss defects. Specifically, after introducing DyDown, precision increased by 0.123, recall improved by 0.204, and

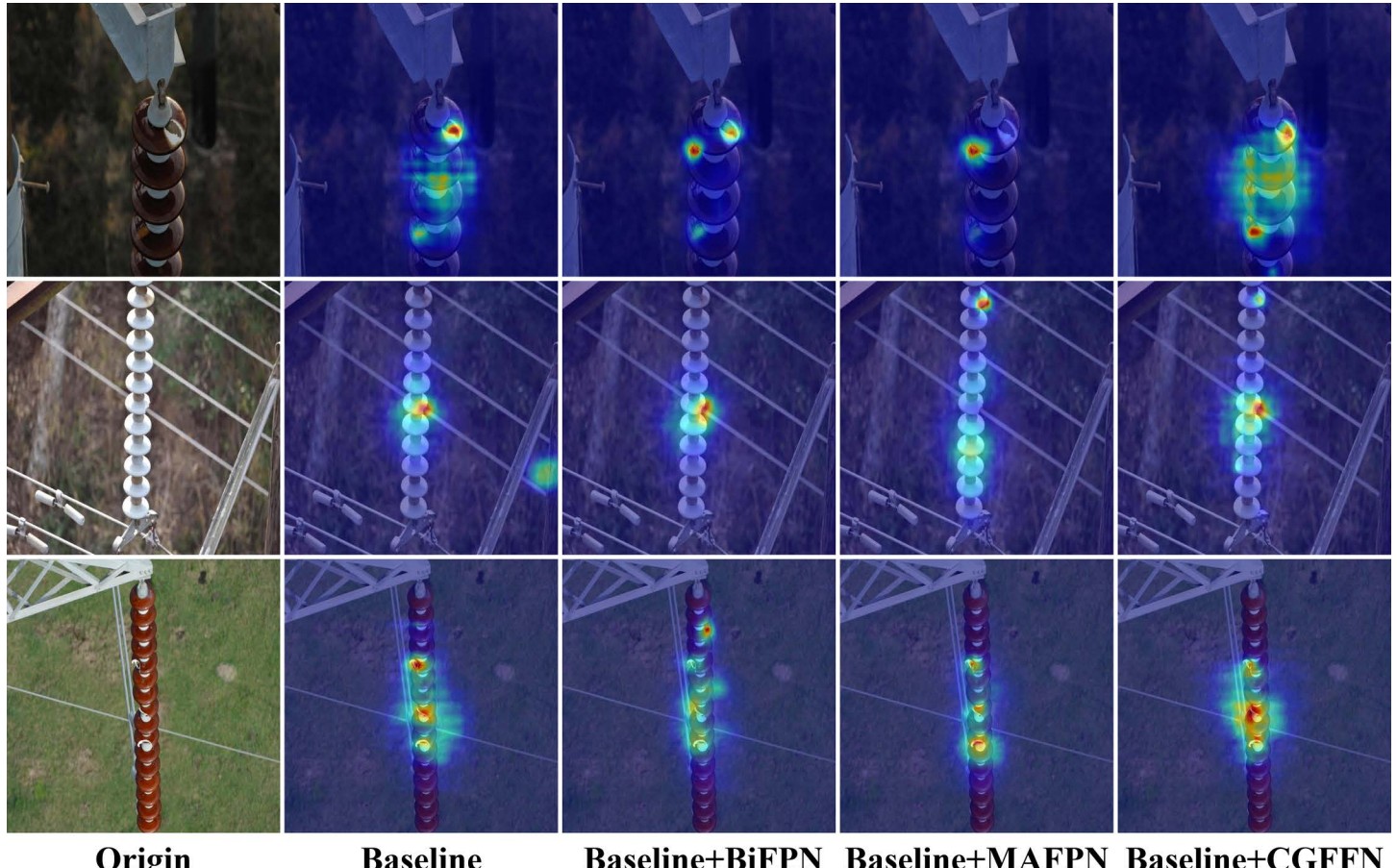

| Origin | Baseline | Baseline+BiFPN | Baseline+MAFPN | Baseline+CGFFN |

**Fig 11. Visual analysis of heat maps on Insulator-DET dataset for baseline models with different feature fusion networks.**

mAP50 increased by 21.1%. This improvement stems from DyDown's capability to extract multi-scale features, particularly benefiting the detection of glass-loss defects. Through its unique dynamic downsampling strategy, DyDown effectively preserves fine-grained details, especially at defect boundaries. This mechanism enhances the model's ability to capture intricate details, which is particularly beneficial for detecting small-scale defects in complex backgrounds. Compared to conventional downsampling methods, DyDown not only retains image details but also reduces computational burden, achieving a better balance between precision and recall, ultimately leading to a significant enhancement in detection performance.

However, DyDown did not achieve the same level of improvement for broken disc defects. While recall increased from 0.589 to 0.695 and mAP50 improved by 3.3%, precision dropped significantly from 0.591 to 0.508. This change highlights a potential issue with DyDown. DyDown dynamically processes input features through convolution and pooling strategies, which enhances recall but may lead to an increased false positive rate for broken disc defects, which exhibit high variability and complexity.

As shown in Table 13, this study compared the impact of different downsampling components on performance using the Insulator-DET validation set. These included the downsampling module from YOLOv7 (V7Sampling) and the downsampling module from YOLOv9 (Adown). Compared to the baseline model, introducing DyDown increased mAP50 by 4.6% and recall by 7.9%. The experimental results demonstrate that DyDown effectively balances precision and recall, mitigating the issue of detail loss during the forward propagation process.

**3) Effectiveness of MBPTB:** As shown in Table 11, MBPTB demonstrated significant performance improvements in detecting glass-loss defects. Specifically, precision and recall increased by 0.057 and 0.083, respectively, while mAP50 improved by 12.3%. Similarly, for polymer defects, MBPTB also exhibited certain advantages, with precision increasing from 0.422 to 0.568, recall from 0.375 to 0.494, and mAP50 improving by 8%. By performing feature separation and fusion at different stages, MBPTB effectively enhances the model's ability to detect defects such as polymer and glass-loss. However, for polymer-dirty defects, although the recall increased from 0.065 to 0.164, the precision dropped significantly from 1.000 to 0.641, while mAP50 improved by 11.5%. This indicates that while MBPTB enhances the recall for polymer-dirty defects, it comes at the cost of reduced precision. A possible reason for this decline is the high noise level associated with polymer-dirty defects. MBPTB's heavy reliance on multi-branch feature fusion may lead to overfitting to fine details, increasing false positives. Although CGLU provides strong flexibility in multi-scale feature fusion, MBPTB fails to effectively suppress responses to irrelevant regions in complex background environments, leading to a decline in precision. Additionally, MBPTB does not exhibit significant performance improvements for glass-dirty and two-glasses defects, suggesting that it may introduce excessive complexity when handling relatively simple defects, thereby degrading overall model performance. Future improvements may require enhancing MBPTB's ability to suppress noise and irrelevant features to better balance recall and precision.

Additionally, to verify the superiority of MBPTB, a backbone replacement experiment was conducted, with the results shown in Table 14. The backbones involved in the experiment included ConvNeXtV2 [47], EfficientViT [48], FasterNet [49], RevCol [50], HGNetV2, MobileNetV4 [51], RepViT [52], and StarNet [53]. As illustrated in Fig 12, the green areas represent the receptive field sizes of the models. The receptive field of the backbone network incorporating MBPTB is

**Table 13. Compare The Performance Of Different Downsampling Component.**

| Methods | P↑ | R↑ | mAP$_{50}$↑ | Params↓ | FPS↑ |
|---|---|---|---|---|---|
| Baseline | **0.726** | 0.499 | 51.0 | 3.00 | **308** |
| Baseline with V7Sampling | 0.521 | 0.572 | 54.8 | 2.97 | 293 |
| Baseline with ADown | 0.657 | 0.495 | 50.6 | **2.94** | 294 |
| Baseline with DyDown | 0.637 | **0.578** | **55.6** | 3.00 | 268 |

**Table 14. Compare the Detection Performance of Different Backbone Architectures.**

| Methods | P↑ | R↑ | mAP$_{50}$↑ | Params↓ | FPS↑ |
|---|---|---|---|---|---|
| Baseline with MBPTB | **0.701** | 0.539 | **54.1** | 2.85 | 254 |
| Baseline with mobilenetv4 | 0.513 | 0.513 | 46.1 | 5.70 | 320 |
| Baseline with RevCol | 0.649 | 0.523 | 52.2 | 2.27 | 275 |
| Baseline with startnet | 0.607 | **0.549** | 50.6 | **2.21** | 387 |
| Baseline with fasternet | 0.48 | 0.528 | 50.8 | 4.17 | 335 |
| Baseline with repvit | 0.642 | 0.483 | 50.2 | 6.71 | 240 |
| Baseline with convnextv2 | 0.441 | 0.388 | 37.5 | 5.66 | 185 |
| Baseline with HGNetV2 | 0.632 | 0.484 | 52.6 | 2.35 | **407** |
| Baseline with efficientViT | 0.520 | 0.470 | 51.0 | 4.01 | 122 |

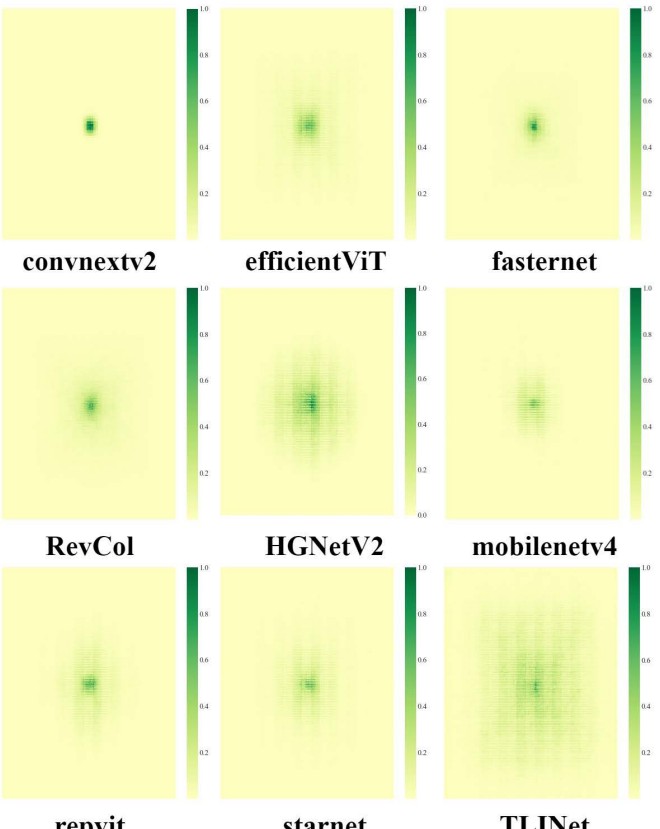

**Fig 12. Visual analysis of receptive fields for baseline models with different backbone networks.**

significantly larger, indicating that MBPTB enhances model performance more effectively than other advanced backbone networks. This further validates the effectiveness of MBPTB.

### 4.7. Generalization verification of our component

To evaluate the generalization capability of the proposed modules, they were integrated with YOLOv5n separately. As shown in Table 15, compared to YOLOv5n, incorporating CGFFN improved the recall by 0.8% and mAP50 by 2.1%.

**Table 15. Performance Comparison of The YOLOv5 Combination with Different Strategies.**

| Method | CGFFN | DyDown | MBPTB | Params↓ | $mAP_{50}$↑ | FPS↑ | P↑ | R↑ |
|--------|-------|--------|-------|---------|-------------|------|-----|-----|
| Baseline | – | – | – | **1.76** | 49.8 | **292** | 0.592 | 0.541 |
| | √ | – | – | 2.34 | 51.9 | 270 | 0.581 | **0.549** |
| | – | √ | – | 2.46 | **53.2** | 263 | **0.678** | 0.52 |
| | – | – | √ | 2.44 | 51.7 | 260 | 0.652 | 0.524 |

Introducing DyDown enhanced the precision by 8.6% and mAP50 by 3.4%. Similarly, integrating MBPTB increased the precision by 6% and mAP50 by 1.9%. The experimental results demonstrate that the proposed modules exhibit excellent compatibility and generalization performance within the YOLO series.

## 5. Conclusion

This paper focuses on the task of object detection in transmission line insulator defect images. To address key challenges such as multi-scale variations, complex background interference, and information loss, we propose an efficient and accurate detection network—TLINet. First, a Context-Guided Feature Focusing Network (CGFFN) is introduced to enhance the model's defect localization capability in cluttered backgrounds by jointly learning contextual information of defects and their surroundings. Second, to improve the detection performance for small-scale defects and reduce computational cost, we design DyDown, which compresses feature maps while preserving fine-grained information, thereby enhancing sensitivity to small objects. Additionally, the proposed MBPTB module leverages local CNNs and global Transformers for collaborative modeling, improving the model's generalization to various defect types.

It is worth noting that all modules in TLINet are designed with a plug-and-play architecture, offering excellent structural generality and flexibility. Users can flexibly combine, replace, or extend these modules according to specific task requirements. This design greatly simplifies customization and deployment across diverse application scenarios, facilitating rapid adaptation to different hardware platforms and detection tasks. Moreover, the modular structure enables efficient upgrades and optimizations—improvements to individual modules can be seamlessly integrated into the overall network, reducing redesign and retraining costs and improving deployment efficiency and maintainability in industrial applications.

On real-world power inspection datasets such as Insulator-DET, CPLID, and IDID, TLINet demonstrates outstanding performance under varying shooting conditions, background complexities, and defect morphologies, showcasing strong cross-scenario adaptability. Furthermore, its excellent performance on the general object detection dataset VOC07 + 12 confirms its robustness and cross-domain generalization across diverse categories and image distributions, laying a solid foundation for real-world industrial deployment.

Considering the variability of UAV deployment environments, we further analyze the model's performance under different lighting and weather conditions. By selecting image subsets captured under sunny and cloudy conditions, we evaluate TLINet's detection accuracy and inference latency. The results show that TLINet exhibits minimal mAP50 fluctuation under different lighting conditions, with a maximum drop of less than 1.2%, indicating strong environmental adaptability and robustness. Additionally, thanks to the lightweight, plug-and-play module design, the inference time remains consistent across environments, with a latency difference of less than 5 milliseconds, meeting the strict real-time requirements of UAV platforms. Brightness perturbation and random occlusion augmentation were introduced during training to improve robustness against lighting changes and partial occlusions. For future work, we plan to incorporate simulation-based enhancements for extreme weather conditions such as rain and snow, aiming to further improve generalization and deployment stability in more complex environments.

Despite TLINet's strong performance in typical defect detection scenarios, certain challenges remain under extreme lighting or severe occlusion. Future efforts will explore illumination-invariant enhancement and adaptive occlusion modeling to further improve robustness in harsh conditions. Adaptive feature fusion and illumination enhancement techniques

can help the model handle lighting fluctuations more reliably, improving real-world detection accuracy. Introducing attention mechanisms and multi-view fusion is expected to mitigate information loss due to occlusion, enhancing the detection of concealed defects. As these challenges are progressively addressed, TLINet will offer greater scalability and practicality, paving the way for broader applications in intelligent power inspection and industrial vision systems.

## Author contributions

**Conceptualization:** Xun Li, Baoxi Yuan.

**Funding acquisition:** Yuzhen Zhao.

**Investigation:** Xiangke Jiao.

**Methodology:** Xun Li.

**Resources:** Yang Zhao, Zhun Guo.

**Software:** Xun Li.

**Supervision:** Yang Zhao.

**Validation:** Yang Zhao.

**Visualization:** Zhun Guo, Yongming Zhang.

**Writing – original draft:** Xun Li.

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
