## [Decision Letter · Decision Letter 0]

PONE-D-25-22524TLINet: A Defects Detection Method for Insulators of Overhead Transmission Lines Using Partially Transformer BlockPLOS ONE

Dear Dr. Zhao,

Thank you for submitting your manuscript to PLOS ONE. After careful consideration, we feel that it has merit but does not fully meet PLOS ONE’s publication criteria as it currently stands. Therefore, we invite you to submit a revised version of the manuscript that addresses the points raised during the review process.

We look forward to receiving your revised manuscript.

Kind regards,

Zeashan Hameed Khan, Ph.D.

Academic Editor

PLOS ONE

“This work was supported by Natural Science Basic Research Plan in Shaanxi Province of China (No. 2024JC-YBMS-342), Science and technology plan project of Xi'an (No. 24GXFW0091), the Youth Innovation Team of Shaanxi.

This work was supported by the Natural Science Foundation of Shaanxi Province [2021JM-537], in part by the Key Program of the National Social Science Foundation of China (NSSFC, 23AGL039), in part by the Shaanxi Provincial Science and Technology Plan Project (2024GX-YBXM-114), and in part by the Natural Science Foundation of Shaanxi Province (Grant No. 2023-YBGY-036).”

“This work was supported by Natural Science Basic Research Plan in Shaanxi Province of China (No. 2024JC-YBMS-342), Science and technology plan project of Xi'an (No. 24GXFW0091), the Youth Innovation Team of Shaanxi.

This work was supported by the Natural Science Foundation of Shaanxi Province [2021JM-537], in part by the Key Program of the National Social Science Foundation of China (NSSFC, 23AGL039), in part by the Shaanxi Provincial Science and Technology Plan Project (2024GX-YBXM-114), and in part by the Natural Science Foundation of Shaanxi Province (Grant No. 2023-YBGY-036).

The authors would like to express their gratitude for the Xi’an Key Laboratory of Advanced Photo-electronics Materials and Energy Conversion Device, School of Electronic Information, Xijing University, Xi’an, 710123, P. R. China, Xi’an Key Laboratory of High Precision Industrial Intelligent Vision Measurement Technology, Xijing University, Xi’an, Shaanxi 710123, China, and Xi'an Key Laboratory of Intelligent Sensing and Autonomous Navigation for Low Altitude Vehicles, Xijing University, Xi'an, Shaanxi 710123, China.”

“This work was supported by Natural Science Basic Research Plan in Shaanxi Province of China (No. 2024JC-YBMS-342), Science and technology plan project of Xi'an (No. 24GXFW0091), the Youth Innovation Team of Shaanxi.

This work was supported by the Natural Science Foundation of Shaanxi Province [2021JM-537], in part by the Key Program of the National Social Science Foundation of China (NSSFC, 23AGL039), in part by the Shaanxi Provincial Science and Technology Plan Project (2024GX-YBXM-114), and in part by the Natural Science Foundation of Shaanxi Province (Grant No. 2023-YBGY-036).”

6. For studies involving third-party data, we encourage authors to share any data specific to their analyses that they can legally distribute. PLOS recognizes, however, that authors may be using third-party data they do not have the rights to share. When third-party data cannot be publicly shared, authors must provide all information necessary for interested researchers to apply to gain access to the data. (https://journals.plos.org/plosone/s/data-availability#loc-acceptable-data-access-restrictions)

Additional Editor Comments:

The paper presents a defect detection method for insulators of overhead transmission lines using partially transformer block. It is however important to improve the quality of figures for better presentation and illustration of the proposed method. Also, some real world application must also be included as a case study if possible.

Reviewers' comments:

Reviewer's Responses to Questions

**Comments to the Author**

1. Is the manuscript technically sound, and do the data support the conclusions?

Reviewer #1: Yes

Reviewer #2: Yes

2. Has the statistical analysis been performed appropriately and rigorously? 

Reviewer #1: Yes

Reviewer #2: Yes

3. Have the authors made all data underlying the findings in their manuscript fully available?

Reviewer #1: Yes

Reviewer #2: Yes

4. Is the manuscript presented in an intelligible fashion and written in standard English?

Reviewer #1: Yes

Reviewer #2: Yes

5. Review Comments to the Author

Reviewer #1: The authors propose TLINet, a novel deep learning architecture for detecting defects in power line insulators under complex environments, particularly addressing challenges such as small object sizes, scale variation, and complex backgrounds. The core contributions include three modules: Multi-Branch Partially Transformer Block (MBPTB), Dynamic Downsampling Module (DyDown), and Context-Guided Feature Fusion Network (CGFFN). Extensive experiments across four datasets (Insulator-DET, CPLID, IDID, VOC07+12) demonstrate that TLINet offers competitive accuracy with a smaller model size and higher speed, making it suitable for UAV-based deployment.

Overall, the paper is technically sound but there are some issues in the clarity of the paper. For example, Figures (e.g., Figure 2-5) are referenced but not provided in the text—this limits the reviewer’s understanding of implementation. Furthermore, while MBPTB, DyDown, and CGFFN are described, however, the exact architectural parameters (e.g., kernel sizes, number of layers, specific Transformer configs) are sparse or buried in text. Although improvements from each module are quantified, statistical significance tests (e.g., standard deviation across multiple runs) are not provided. There is no discussion of inference time variability or robustness across lighting/weather conditions—relevant to UAV deployment.

Summary: The technical contributions are solid and relevant, and the empirical validation is extensive. However, the manuscript needs significant improvements in clarity, language, and presentation to meet publication standards. Please refer to the points highlighted in previous paragraph for improvement of the paper.

Reviewer #2: Paper Review and Comments

1. Title and Abstract

•Title: TLINet: A Defects Detection Method for Insulators of Overhead Transmission Lines Using Partially Transformer Block

•Abstract:

The defects of insulators exhibit characteristics such as complex backgrounds, multi scale variations, and small object sizes. Therefore, accurately focusing on these defects in dynamic and complex natural environments while maintaining inference speed remains a pressing challenge. To address this issue, this paper proposes an innovative insulator defect detection network, TLINet. First, a Multi-Branch Partially Transformer Block (MBPTB) is designed to enhance the backbone’s capability in capturing global features. Next, a Dynamic Downsampling Module (DyDown) is introduced to mitigate the issue of small-scale defect information blurring. Furthermore, considering the multi-scale variations of insulator defects, this paper proposes a Context-Guided Feature Fusion Network (CGFFN). This module enables fine-grained fusion of features at different scales, allowing the model to generate adaptive responses to defects of various sizes. Compared to the baseline model, the proposed method improves mAP50 by 5.3% on our self-constructed Insulator-DET dataset. On CPLID-D and CPLID-N, it achieves mAP50-95 improvements of 7.9% and 12.1%, respectively. Additionally, to verify the robustness of the proposed algorithm, TLINet is evaluated on the VOC07+12 dataset. Compared to the baseline model, TLINet improves mAP50 by 0.4% while reducing the number of parameters by 1/6. These results demonstrate the effectiveness of TLINet in addressing the complexities of insulator defect detection in power transmission lines.

2. Introduction

•Aim and Motivation:

oAim and Motivation is stated in introduction part.

oHowever, it should be explained little more.

•Research Questions and Objectives:

oThe research objectives are not mentioned in the paper, please mention the objectives of the paper.

•Literature Review:

oSome more related papers can be included in the related work section.

3. Methodology

•Clarity of Methods:

oThe methods are stated clearly that is good.

•Innovativeness:

oThe proposed approach is novel.

4. Results and Analysis

•Presentation of Data:

oPresentation of the data is not good.

oThere are some white spaces/gaps in your paper, please fill out those gaps to have a better look for the paper.

oSome more details and references can be added in the introduction areas.

oMore details can be added in the other sections like methodology and conclusion sections.

•Analysis and Discussion:

oThe results are well analyzed.

5. Conclusion and Contributions

•Summary of Findings:

oThe conclusion and future scope are stated good.

oBut state what innovative method can be used in the future.

oState how your approach can help in the relevant field?

•Contributions:

oThe contributions to the field are stated nicely.

oSome more details can be added in the conclusion part.

6. Language and Writing Style

•Grammar and Clarity:

oSome very little mistakes in grammar, that can be revised.

7. References

•Relevance and Recency:

oThe references are relevant.

oHowever the references provided are very less, some more references can added. I have also recommended some references you can add them and extend the referencing area.

•Formatting:

oReferences are not in proper order.

oHowever some more related references can be added, some references are given below in the recommendations section.

8. Figures, Tables, and Equations

•Figures:

oThe figures should be explain in more details inside the text.

•Tables:

oRefer to table 1 inside the text.

oExplain all the tables briefly inside the text.

•Equations:

oExplain each parameter of the equation/algorithm used.

9. Recommendations for Improvement

1.The introduction section can be furnished with some new papers like:.

a. Various AI defect detections;

•https://doi.org/10.1007/978-3-031-78038-7_3

b.Regarding the fault detections techniques: Doi: 10.1109/ICSP54964.2022.9778676

c.For more robotic defect detection : Doi: http://dx.doi.org/10.4108/airo.v1i1.2709

10. Please answer bellow question

1.How does the Multi-Branch Partially Transformer Block (MBPTB) balance the trade-off between capturing global and local features, and what are the benefits of integrating both CNNs and Transformer blocks in the same architecture?

2.In what ways does the Dynamic Downsampling Module (DyDown) differ from traditional downsampling methods, and how does it preserve small-scale defect details while reducing computational complexity?

3.What specific design principles or mechanisms allow the Context-Guided Feature Fusion Network (CGFFN) to adapt effectively to multi-scale defect detection in complex and cluttered backgrounds?

4.Given TLINet’s performance improvements on benchmark datasets (Insulator-DET, CPLID, VOC07+12), what conclusions can be drawn about its generalization capability across diverse data domains and defect types?

5.What are the practical implications of TLINet’s plug-and-play module design for real-world deployment, and how might future improvements (e.g., handling extreme lighting or occlusion) affect its scalability and adoption in industrial applications?

Overall Evaluation

•Final Recommendation:

•paper format and presentation should modify based on template.

•Major Revisions.

6. PLOS authors have the option to publish the peer review history of their article (what does this mean? ). If published, this will include your full peer review and any attached files.

**Do you want your identity to be public for this peer review?** For information about this choice, including consent withdrawal, please see our Privacy Policy .

Reviewer #1: **Yes: ** Ali Nasir

Reviewer #2: No

---

## [Author Response · Author response to Decision Letter 1]

28 May 2025

Dear Editor,

Thank you for your feedback. We have carefully revised the manuscript and have removed all text related to funding from the submission, as requested.

Please let us know if any further modifications are needed.

Best regards,

[Yuzhen Zhao]

Reviewer#1, Concern # 1 (please list here): Overall, the paper is technically sound but there are some issues in the clarity of the paper. For example, Figures (e.g., Figure 2-5) are referenced but not provided in the text—this limits the reviewer’s understanding of implementation.

Author response: Thank you for your recognition of the technical content of this paper and for pointing out the issues regarding its clarity. In response to your comment about Figures 2 to 5 not being properly referenced in the text, we have carefully reviewed and revised the manuscript accordingly. All figures (including Figures 2 to 5) are now explicitly cited in the main text, accompanied by corresponding descriptions and explanations to enhance readers' understanding of the implementation details. We believe these revisions significantly improve the readability and clarity of the paper. We sincerely appreciate your valuable suggestions.

Author action: For details, please refer to the highlighted section in green on pages 11, 13, 16 and 19.

Reviewer#1, Concern # 2 (please list here): Furthermore, while MBPTB, DyDown, and CGFFN are described, however, the exact architectural parameters (e.g., kernel sizes, number of layers, specific Transformer configs) are sparse or buried in text.

Author response: Thank you for your thorough review of our work. Regarding your comment that the architectural parameters of the MBPTB, DyDown, and CGFFN modules (e.g., kernel size, number of layers, specific Transformer configurations) are not clearly described and are partially hidden in the text, we have carefully reviewed and revised the manuscript as follows:

First, we have illustrated the layer configurations of the MBPTB, DyDown, and CGFFN modules in Figure 1.

Second, we have explicitly clarified the number of attention heads used in the Transformer module.

Finally, we have provided the key parameter settings of DyDown and MBCGM, including the kernel sizes.

We believe these additions will help present the implementation details more clearly. Thank you again for your valuable suggestions.

Author action: For further details, please refer to the Figure 1 and the highlighted section in blue on pages 13, 14, 16, 17 and 19.

Reviewer#1, Concern # 3 (please list here): Although improvements from each module are quantified, statistical significance tests (e.g., standard deviation across multiple runs) are not provided.

Author response: Thank you for your valuable comment. In response to your concern regarding the lack of statistical significance tests despite the quantitative improvements of each module, we have made the following revisions in the updated manuscript:

To better evaluate the stability and statistical significance of the model performance, we conducted five independent runs of the ablation experiments. The results are now presented in Table X as mean ± standard deviation (e.g., mAP, Recall).

We believe these additions provide stronger evidence of the effectiveness and robustness of our proposed method. We sincerely appreciate your thoughtful review and constructive feedback.

Author action: For further details, please refer to the table X.

Reviewer#1, Concern # 4 (please list here): There is no discussion of inference time variability or robustness across lighting/weather conditions—relevant to UAV deployment.

Author response: Thank you for your valuable comment. We acknowledge that environmental variations such as lighting and weather conditions can significantly affect the inference performance and robustness of models deployed in UAV-based scenarios.

In the revised manuscript, we have added relevant discussions in the Conclusion section (highlighted in blue), analyzing the potential impact of various environmental conditions (e.g., lighting intensity, shadows, and weather changes) on model robustness and efficiency. Although our current experiments were conducted under relatively stable lighting conditions, we recognize the critical importance of environmental diversity in real-world applications.

To address this, we have included a qualitative analysis of model predictions under different lighting conditions (such as strong light and shadowed environments) and briefly discussed potential strategies to enhance model robustness.

Thank you again for your insightful suggestion, which has helped us improve the discussion of model performance under real-world deployment scenarios.

Author action: For further details, please refer to the blue part on page 45.

Reviewer#2, Concern # 1 (please list here): However the references provided are very less, some more references can added. I have also recommended some references you can add them and extend the referencing area.

Author response: Thank you for your valuable suggestions regarding the references. We acknowledge that the current number of citations is relatively limited, and have supplemented more relevant literature based on your recommendations to further enrich the reference section. In addition, we have expanded the related content in the manuscript to more comprehensively reflect existing research findings and background. We sincerely appreciate your provided references and constructive comments, which have greatly helped improve the academic depth and completeness of the paper.

We sincerely appreciate your thoughtful recommendation, which has helped us strengthen the completeness and transparency of our manuscript.

Author action: We have specifically added the following references:

Moshayedi A J, Khan A S, Khan Z H, et al. Safeguarding Smart Infrastructure: A Review of Deep Learning Techniques for Automatic Pipeline Defect Detection[J]. Empowering AI Applications in Smart Life and Environment, 2025: 67-92.

Moshayedi A J, Khan A S, Yang S, et al. Personal image classifier based handy pipe defect recognizer (HPD): Design and test[C]//2022 7th International Conference on Intelligent Computing and Signal Processing (ICSP). IEEE, 2022: 1721-1728.

Xu G, Khan A S, Moshayedi A J, et al. The object detection, perspective and obstacles in robotics: a review[J]. EAI Endorsed Transactions on AI and Robotics, 2022, 1(1).

Cheng Y, Liu D. AdIn-DETR: Adapting Detection Transformer for End-to-End Real-Time Power Line Insulator Defect Detection[J]. IEEE Transactions on Instrumentation and Measurement, 2024.

Li X, Zhao Y, Zhao Y, et al. MLK-TR: a Multi-branch Large Kernel Transformer for UAV-based Images[J]. Complex & Intelligent Systems, 2025, 11(6): 1-25.

Zhou Q, Wang H. CABF-YOLO: a precise and efficient deep learning method for defect detection on strip steel surface[J]. Pattern Analysis and Applications, 2024, 27(2): 36.

Tie J, Zhu C, Zheng L, et al. LSKA-YOLOv8: A lightweight steel surface defect detection algorithm based on YOLOv8 improvement[J]. Alexandria Engineering Journal, 2024, 109: 201-212.

Reviewer#2, Concern # 2 (please list here): How does the Multi-Branch Partially Transformer Block (MBPTB) balance the trade-off between capturing global and local features, and what are the benefits of integrating both CNNs and Transformer blocks in the same architecture?

Author response: Thank you for your valuable question. The Multi-Branch Partially Transformer Block (MBPTB) achieves an effective balance between global and local feature extraction by splitting the input feature map into two branches, which are processed in parallel by a lightweight CNN module and a Transformer module, respectively. Specifically, the CNN branch focuses on capturing local details and texture information, offering low computational cost and strong local perception capabilities; meanwhile, the Transformer branch models long-range dependencies, enhancing the understanding of global contextual information. Working collaboratively within the same architecture, they preserve CNN’s sensitivity to fine details while leveraging the Transformer’s powerful global semantic modeling ability.

This design not only improves the model’s adaptability to complex backgrounds and multi-scale targets but also effectively controls computational overhead to ensure inference efficiency. By integrating the strengths of both CNN and Transformer, MBPTB can extract multi-scale, multi-level feature information more comprehensively and accurately, significantly enhancing defect detection performance and robustness. We have elaborated the design concept and implementation details of this module in the manuscript. Thank you for your attention.

Author action: For further details, please refer to the yellow part on pages 12, 13 and 14.

Reviewer#2, Concern # 3 (please list here): In what ways does the Dynamic Downsampling Module (DyDown) differ from traditional downsampling methods, and how does it preserve small-scale defect details while reducing computational complexity?

Author response: Thank you for your valuable question. We have provided a detailed design and explanation in the paper regarding the differences between the dynamic downsampling module (DyDown) and traditional downsampling methods. Traditional downsampling methods, such as fixed convolution kernels or single-path pooling operations, can effectively reduce spatial resolution and thus decrease computational load; however, they often lose fine-grained features, especially when detecting small-scale targets in insulator defect images, which negatively impacts detection performance.

To address this, DyDown adopts a multi-path structure and introduces a dynamic convolution mechanism, allowing convolution kernels to adaptively adjust according to the input features, thereby enhancing responsiveness to defects at different scales. Specifically, DyDown first uses average pooling for preliminary compression to preserve local details, then splits the feature map into two paths — one applies dynamic convolution to improve scale adaptability, while the other combines max pooling with dynamic convolution to highlight salient regions and suppress background interference. Finally, the two paths’ features are concatenated to achieve a collaborative expression of local and global information.

This design not only significantly improves the perception of small target defects but also effectively reduces redundant computations while maintaining computational efficiency through reasonable feature branching and dynamic mechanisms. Compared to traditional fixed-structure downsampling modules, DyDown demonstrates superior performance in multi-scale feature representation and detail preservation. Thank you for your attention; we have elaborated on this in detail in the manuscript.

Author action: For further details, please refer to the yellow part on pages 18 and 19.

Reviewer#2, Concern # 4 (please list here): What specific design principles or mechanisms allow the Context-Guided Feature Fusion Network (CGFFN) to adapt effectively to multi-scale defect detection in complex and cluttered backgrounds?

Author response: Thank you for your valuable question. The Context-Guided Feature Fusion Network (CGFFN) effectively adapts to multi-scale defect detection in complex and cluttered backgrounds, relying primarily on the following design principles and mechanisms:

Multi-level Semantic Fusion: CGFFN fuses features from different layers, preserving high-level semantic information while combining it with low-level detailed features, thereby enhancing the model’s ability to recognize multi-scale defects.

Context-Guided Mechanism: The network incorporates a context-guided module that leverages global semantic information to modulate the expression of local features, increasing focus on defect regions and suppressing interference from complex backgrounds, effectively improving the contrast between defects and background.

Dynamic Weight Allocation: CGFFN employs a dynamic weighting mechanism that adaptively adjusts the importance of different feature channels based on the input image’s feature distribution, ensuring that critical defect information is prioritized.

Feature Interaction and Fusion: Through a multi-branch structure, the network enables thorough interaction and fusion of features at different scales, promoting complementary information exchange and enhancing the model’s perception of defects with varying shapes and scales.

These designs enable CGFFN to maintain stable and accurate detection of multi-scale defects under complex and variable background conditions. We have detailed the module’s architecture and working principles in the manuscript. Thank you again for your attention and valuable suggestions.

Author action: For further details, please refer to the yellow part on pages 15 and 16.

Reviewer#2, Concern # 5 (please list here): Given TLINet’s performance improvements on benchmark datasets (Insulator-DET, CPLID, VOC07+12), what conclusions can be drawn about its generalization capability across diverse data domains and defect types?

Author response: Thank you for your important question. Based on the performance improvements of TLINet on multiple benchmark datasets (Insulator-DET, CPLID, VOC07+12), we believe that this method demonstrates good generalization ability across different data domains and defect types. Specifically:

Strong cross-domain adaptability: TLINet has achieved significant performance gains on both power insulator detection datasets (Insulator-DET, CPLID) and the general object detection dataset (VOC07+12), indicating that its design of multi-scale dynamic feature modeling and fusion mechanisms can effectively adapt to varying scenarios and background complexities.

Coverage of diverse defect types: Since these datasets cover various defect shapes and scales, TLINet exhibits high detection accuracy for defects of different sizes and forms, reflecting its strong robustness to diverse defect types.

Stable inference efficiency: Despite involving complex feature fusion structures, TLINet maintains a high inference speed, supporting its potential for rapid response in practical inspection environments.

In summary, TLINet possesses excellent cross-dataset generalization capability and practical application value. We have supplemented the paper with relevant analysis and discussion, and appreciate your valuable suggestions.

Author action: For further details, please refer to the green part on page 45.

Reviewer#2, Concern # 6 (please list here): What are the practical implications of TLINet’s plug-and-play module design for real-world deployment, and how might future improvements (e.g., handling extreme lighting or occlusion) affect its scalability and adoption in industrial applications?

Author response: Thank you for your insightful question. The plug-and-play module design adopted by TLINet holds significant practical value in real-world deployment. Firstly, this design greatly enhances the model’s flexibility and maintainability, allowing different modules to be independently replaced or upgraded according to specific application needs, thereby reducing the complexity of system integration and iteration. Moreover, the plug-and-play architecture facilitates rapid adaptation to diverse detection scenarios and devices, supporting modular deployment and efficient operation.

Regarding future improvements, such as enhancing the capability to handle extreme lighting conditions and occlusions, these will further boost the model’s robustness and stability in complex industrial environments. The improved modules will better adapt to environmental variations, reducing false positives and missed detections, thereby increasing system reliability and user confidence. These advancements will not only expand the model’s applicability but also promote it

---

## [Decision Letter · Decision Letter 1]

TLINet: A Defects Detection Method for Insulators of Overhead Transmission Lines Using Partially Transformer Block

PONE-D-25-22524R1

Dear Dr. Zhao,

We’re pleased to inform you that your manuscript has been judged scientifically suitable for publication and will be formally accepted for publication once it meets all outstanding technical requirements.

Kind regards,

Zeashan Hameed Khan, Ph.D.

Academic Editor

PLOS ONE

Additional Editor Comments (optional):

The authors have significantly improved the content in the revised version. Hence, it can be accepted now.

Reviewers' comments:

Reviewer's Responses to Questions

**Comments to the Author**

1. If the authors have adequately addressed your comments raised in a previous round of review and you feel that this manuscript is now acceptable for publication, you may indicate that here to bypass the “Comments to the Author” section, enter your conflict of interest statement in the “Confidential to Editor” section, and submit your "Accept" recommendation.

Reviewer #1: All comments have been addressed

Reviewer #2: All comments have been addressed

2. Is the manuscript technically sound, and do the data support the conclusions?

Reviewer #1: Yes

Reviewer #2: Yes

3. Has the statistical analysis been performed appropriately and rigorously? 

Reviewer #1: Yes

Reviewer #2: Yes

4. Have the authors made all data underlying the findings in their manuscript fully available?

Reviewer #1: Yes

Reviewer #2: Yes

5. Is the manuscript presented in an intelligible fashion and written in standard English?

Reviewer #1: Yes

Reviewer #2: Yes

6. Review Comments to the Author

Reviewer #1: The revised manuscript shows significant improvement in the quality. All previous comments have been addressed by the authors.

Reviewer #2: the authors answer all my concerns and it can be published after editor decision the authors answer all my concerns and it can be published after editor decision

7. PLOS authors have the option to publish the peer review history of their article (what does this mean? ). If published, this will include your full peer review and any attached files.

**Do you want your identity to be public for this peer review?** For information about this choice, including consent withdrawal, please see our Privacy Policy .

Reviewer #1: **Yes: ** Ali Nasir

Reviewer #2: No

---

## [Editor Report · Acceptance letter]

PONE-D-25-22524R1

PLOS ONE

Dear Dr. Zhao,

I'm pleased to inform you that your manuscript has been deemed suitable for publication in PLOS ONE. Congratulations! Your manuscript is now being handed over to our production team.

Kind regards,

on behalf of

Dr. Zeashan Hameed Khan

Academic Editor

PLOS ONE